# Belief-Based Offline Reinforcement Learning for Delay-Robust Policy Optimization

**Simon Sinong Zhan**[1]* **Qingyuan Wu**[2]* **Philip Wang**[1] **Frank Yang**[1] **Xiangyu Shi**[1]
**Chao Huang**[2] **Qi Zhu**[1]

[1]Northwestern University    [2]University of Southampton

## Abstract

Offline-to-online deployment of reinforcement learning (RL) agents often stumbles over two fundamental gaps: (1) the *sim-to-real gap*, where real-world systems exhibit latency and other physical imperfections not captured in simulation; and (2) the *interaction gap*, where policies trained purely offline face out-of-distribution (OOD) issues during online execution, as collecting new interaction data is costly or risky. As a result, agents must generalize from static, delay-free datasets into dynamic, delay-prone environments. In this work, we propose **DT-CORL** (**D**elay-**T**ransformer belief policy **C**onstrained **O**ffline **RL**), a novel framework for learning delay-robust policies solely from static, delay-free offline data. DT-CORL introduces a transformer-based belief model to infer latent states from delayed observations and jointly trains this belief with a constrained policy objective, ensuring that value estimation and belief representation remain aligned throughout learning. Crucially, our method does not require access to delayed transitions during training and outperforms naive history-augmented baselines, state-of-the-art delayed RL methods, and existing belief-based approaches. Empirically, we demonstrate that DT-CORL achieves strong delay-robust generalization across both locomotion and goal-conditioned tasks in the D4RL benchmark under varying delay regimes. Our results highlight that joint belief-policy optimization is essential for bridging the sim-to-real latency gap and achieving stable performance in delayed environments.

## 1 Introduction

Real-world autonomy, ranging from embodied assistants to autonomous robots (Jiao et al., 2024; Wang et al., 2023a;b; Wei et al., 2017; Zhan et al., 2024), routinely faces *observation and actuation delays*, where sensing lags and control commands arrive late due to computation or communication (Cao et al., 2020; Mahmood et al., 2018; Sun et al., 2022). Such delays violate the Markov assumption and cause severe performance degeneration. Existing solutions often approximate the problem with Delayed Differential Equations (DDEs) (Bellen & Zennaro, 2013; Zhu et al., 2021) or augmented MDPs (Altman & Nain, 1992).

Collecting *online* interaction data under delayed dynamics is unsafe, costly, or impractical (Li et al., 2023; Yang et al., 2024; Zou et al., 2015). In contrast, many systems already provide large *delay-free* datasets from simulators or legacy controllers (Fu et al., 2020; Mu et al., 2021), which omit deployment-time latencies. This mismatch is especially pronounced in various applications: simulations and idealized hardware logs (Makoviychuk et al., 2021; Krishnan et al., 2021) are delay-free, whereas real platforms, such as drones, warehouse manipulators, cloud-robot systems, or even high-frequency trading pipelines, inevitably suffer delays from computation and communication (Gupta & Chow, 2009; Tang et al., 2025). Since collecting trajectories with realistic delays is infeasible, the key challenge is to leverage delay-free data while ensuring robustness to delayed dynamics at deployment. Consequently, we investigate the following question:

---

*Equal contribution, Contact Emails: SinongZhan2028@u.northwestern.edu, qzhu@northwestern.edu

> **How can we train a robust policy *offline*—using only pre-collected, delay-free trajectories—so that it performs reliably when executed in a *delayed* real-world environment?**

This question extends to two pressing needs: (i) leveraging static offline data without further interaction with the environment (offline RL), and (ii) robustly coping with latency at online deployment (delayed RL). Achieving both simultaneously promises safer development cycles, reduced sample complexity, and smoother sim-to-real transfer for latency-sensitive robotic and autonomous systems.

Delayed RL arises when perceptual or actuation latencies break the Markov property, forcing agents to reason over the gap between true system states and delayed observations. Existing solutions fall into two categories. **State augmentation** methods (Kim et al., 2023; Liotet et al., 2022; Wu et al., 2024b;a) extend the state with stacked histories, but incur severe sample inefficiency as dimensionality grows. **Belief-state methods** (Chen et al., 2021; Karamzade et al., 2024; Wu et al., 2025) compress histories into latent beliefs, yet are prone to compounding prediction errors and distribution mismatch when deployed. In parallel, **offline RL** (Levine et al., 2020) trains policies from static datasets without online interaction, but struggles to balance conservatism against effective generalization, often yielding cautious or suboptimal behavior. When delay handling is combined with offline training, these difficulties intensify. Naive augmentation produces artificial delay distributions that are poorly covered by static data, leading to extreme sample inefficiency and dimensional blow-up. Meanwhile, policies and belief models trained separately offline lack corrective feedback: at deployment, even minor distribution shifts force the policy to query unseen state–action pairs, where the frozen belief module must extrapolate. Errors then accumulate step by step, compounding over long horizons and causing rapid performance collapse.

In this work, we propose **DT-CORL** (**D**elay-aware **T**ransformer-belief **C**onstrained **O**ffline **RL**), a novel belief-based framework explicitly designed to tackle the compounded challenges arising in the offline delayed RL setting. By employing a transformer to infer compact belief states, DT-CORL reformulates policy-constrained offline learning in delayed MDPs as standard MDP optimization, enabling delay-robust policies to be trained directly from delay-free data. This end-to-end formulation both improves sample efficiency and belief model expectation during training, thereby sidestepping the distribution-mismatch and compounding-error issues that plague original belief-state pipelines. Empirically, comprehensive experiments on the D4RL suite (Fu et al., 2020) across various delay scenarios—including varying delay lengths, deterministic delays, and stochastic delays—consistently confirm the superiority of DT-CORL over baseline methods, encompassing SOTA online delayed RL methods, augmented-state approaches, and belief-based methods (i.e., direct integration of pretrained belief models with offline RL algorithms).

In Sec. 2, we review related literature. In Sec. 3, we introduce the necessary background and assumptions on delayed Markov decision processes (MDPs). In Eq. (1), we explicitly define the offline delayed RL problem and the corresponding policy-iteration constrained optimization formulation in the augmented delayed MDP. In Sec. 4.1, we provide theoretical connection between the above policy optimization formulation and belief-based constrained policy iteration. Then, in Sec. 4.2, we present our proposed DT-CORL framework, illustrating practical implementation details of our algorithm. Comprehensive empirical evaluations are presented in Sec. 5. Finally, we conclude with a discussion of findings and implications in Sec. 6.

## 2 RELATED WORK

**Delayed RL.** Delayed reinforcement learning arises in domains such as high-frequency trading (Hasbrouck & Saar, 2013) and transportation (Cao et al., 2020). While reward delay has been extensively analyzed (Arjona-Medina et al., 2019; Han et al., 2022; Zhang et al., 2023), we focus on the more challenging *observation/action* delay. Existing solutions follow two main strategies. **Augmentation-based methods** restore the Markov property by stacking the past $\Delta$ actions (and sometimes states), then learning policies in the enlarged state space. Examples include DIDA (Liotet et al., 2022), DC/AC (Bouteiller et al., 2020), ADRL and BPQL (Kim et al., 2023; Wu et al., 2024b), which bootstrap from small-delay tasks, and VDPO (Wu et al., 2024a), which frames delayed control as a variational inference problem. Their weakness is structural: the augmented dimension grows linearly with $\Delta$, causing sample inefficiency and poor scalability. **Belief-based methods** instead compress histories into latent states and act in the original space. DATS performs Gaussian filtering (Chen et al., 2021), D-Dreamer builds recurrent world models (Karamzade et al., 2024), D-SAC

uses causal-transformer attention (Liotet et al., 2021), and DFBT applies sequence-to-sequence transformers to reduce compounding error (Wu et al., 2025). These approaches are more compact, but still accumulate belief error over long rollouts and face distribution mismatch when deployed. All of the above methods assume online interaction with a delayed environment. Aside from some imitation-style studies that pre-train on delay-free logs and then fine-tune online (Liotet et al., 2022; Chen et al., 2021), we are unaware of any work that learns delay-robust policies *offline* using only delay-free data.

**Offline RL and Online Adaptation.** Offline reinforcement learning seeks to train high-performing policies from fixed datasets without further interaction (Shi et al., 2026), addressing safety and data-collection constraints in domains such as robotics, healthcare, and recommendation systems (Levine et al., 2020). Early approaches emphasized *value conservatism*—for instance, CQL (Kumar et al., 2020) and IQL (Kostrikov et al., 2021) penalize Q-values of out-of-distribution (OOD) actions to prevent overestimation. Complementary *policy-constraint* methods such as BRAC (Wu et al., 2019) and TD3,+BC (Fujimoto & Gu, 2021) regularize the learned policy toward the behavior policy to remain within dataset support. While these strategies reduce OOD failure, they can still yield suboptimal policies if the dataset lacks trajectories near the optimum. To address this, *offline-to-online* (hybrid) methods bootstrap from an offline policy and fine-tune with limited online data, e.g., AWAC (Nair et al., 2020), Conservative Fine-Tuning (Nakamoto et al., 2023), and Hy-Q (Song et al., 2022). Other work seeks to bridge mismatches between simulators and the real world (Feng et al., 2023; Niu et al., 2022; Tiboni et al., 2023), but none address the equally important challenge of *delay gaps*—when offline data is delay-free while deployment involves delayed dynamics. This challenge echoes classic control theory, where time-delay compensation has been studied for decades. The Smith Predictor (Smith, 1957), for example, optimizes a controller on a delay-free internal model and predicts the "present" plant output to counteract runtime delays (Normey-Rico & Camacho, 2007; Grimholt & Skogestad, 2018). Such techniques are widely used in industry precisely because they leverage delay-free models to operate reliably under delayed dynamics (Thomas et al., 2020; Mejía et al., 2022; Moraes et al., 2024). In a similar spirit, our setting seeks to learn from delay-free offline data while ensuring robustness at deployment in delayed environments.

## 3 PRELIMINARIES AND PROBLEM FORMULATION

**MDP.** An RL problem is typically formulated as a finite-horizon Markov Decision Process (MDP), defined by the tuple $\langle \mathcal{S}, \mathcal{A}, \mathcal{P}, r, H, \gamma, \rho_0 \rangle$(Sutton et al., 1998). Here, $\mathcal{S}$ denotes the state space, $\mathcal{A}$ the action space, $\mathcal{P} : \mathcal{S} \times \mathcal{A} \times \mathcal{S} \to [0, 1]$ the probabilistic transition kernel, $r : \mathcal{S} \times \mathcal{A} \to \mathbb{R}$ the reward function, $H$ the horizon length, $\gamma \in (0, 1)$ the discount factor, and $\rho_0$ the initial state distribution. At each timestep $t$, given the current state $s_t \in \mathcal{S}$, the agent selects an action $a_t \sim \pi(\cdot|s_t)$ according to policy $\pi : \mathcal{S} \times \mathcal{A} \to [0, 1]$. Subsequently, the MDP transitions to the next state $s_{t+1} \sim \mathcal{P}(\cdot|s_t, a_t)$, and the agent receives a scalar reward $r_t := r(s_t, a_t)$. We further introduce several mild assumptions commonly adopted in the RL literature (Liotet et al., 2022; Rachelson & Lagoudakis, 2010):

**Definition 3.1 (Lipschitz Continuous Policy (Rachelson & Lagoudakis, 2010)).** A stationary Markovian policy $\pi$ is $L_\pi$-LC if for all $s_1, s_2 \in \mathcal{S}$,

$$\mathcal{W}_1\left(\pi(\cdot|s_1),\, \pi(\cdot|s_2)\right) \le L_\pi\, d_{\mathcal{S}}(s_1, s_2).$$

**Definition 3.2 (Lipschitz Continuous MDP (Rachelson & Lagoudakis, 2010)).** An MDP is $(L_{\mathcal{P}}, L_{\mathcal{R}})$-Lipschitz Continuous (LC) if for all $(s_1, a_1), (s_2, a_2) \in \mathcal{S} \times \mathcal{A}$,

$$\mathcal{W}_1\left(\mathcal{P}(\cdot|s_1, a_1),\, \mathcal{P}(\cdot|s_2, a_2)\right) \le L_{\mathcal{P}}\left(d_{\mathcal{S}}(s_1, s_2) + d_{\mathcal{A}}(a_1, a_2)\right),$$

$$|r(s_1, a_1) - r(s_2, a_2)| \le L_{\mathcal{R}}\left(d_{\mathcal{S}}(s_1, s_2) + d_{\mathcal{A}}(a_1, a_2)\right),$$

where $\mathcal{W}_1$ is L1-Wasserstein distance.

**Definition 3.3 (Lipschitz Continuous Q-function (Rachelson & Lagoudakis, 2010)).** Consider an $(L_{\mathcal{P}}, L_{\mathcal{R}})$-LC MDP and an $L_\pi$-LC policy $\pi$. If the discount factor $\gamma$ satisfies $\gamma L_{\mathcal{P}}(1 + L_\pi) < 1$, then the action-value function $Q^\pi$ is $L_Q$-Lipschitz continuous for some finite constant $L_Q > 0$.

**Delayed MDP.** A delayed RL problem can be reformulated as a delayed MDP with Markov property based on the augmentation approaches (Altman & Nain, 1992). Assuming the delay being $\Delta$, the delayed MDP is denoted as a tuple $\langle \mathcal{X}, \mathcal{A}, \mathcal{P}_\Delta, r_\Delta, H, \gamma, \rho_\Delta \rangle$, where the augmented state

space is defined as $\mathcal{X} := \mathcal{S} \times \mathcal{A}^{\Delta}$ (e.g., an augmented state $x_t = \{s_{t-\Delta}, a_{t-\Delta}, \cdots, a_{t-1}\} \in \mathcal{X}$), $\mathcal{A}$ is the action space, the delayed transition function is defined as $\mathcal{P}_{\Delta}(x_{t+1}|x_t, a_t) := \mathcal{P}(s_{t-\Delta+1}|s_{t-\Delta}, a_{t-\Delta})\delta_{a_t}(a_t')\prod_{i=1}^{\Delta-1}\delta_{a_{t-i}}(a_{t-i}')$ where $\delta$ is the Dirac distribution, the delayed reward function is defined as $r_{\Delta}(x_t, a_t) := \mathbb{E}_{s_t \sim b(\cdot|x_t)}[r(s_t, a_t)]$ where $b$ is the belief function defined as $b_{\Delta}(s_t|x_t) := \int_{\mathcal{S}^{\Delta}}\prod_{i=0}^{\Delta-1}\mathcal{P}(s_{t-\Delta+i+1}|s_{t-\Delta+i}, a_{t-\Delta+i})\mathrm{d}s_{t-\Delta+i+1}$, the initial augmented state distribution is defined as $\rho_{\Delta} = \rho\prod_{i=1}^{\Delta}\delta_{a_{-i}}$. Noted that delayed RL is not necessarily only observational delay, there could be action delay as well. However, it has been proved that action-delay formulated delay RL problem is a subset of observation-delay RL problem (Katsikopoulos & Engelbrecht, 2003). Thus, without loss of generality, we only consider the general observation delay in the remainder of the paper.

**Problem Formulation.** We consider the offline delayed RL setting in which the agent learns solely from a pre-collected static dataset, $\mathcal{D} = \{(s_t^i, a_t^i, r_t^i, s_{t+1}^i)\}_{t=0}^{H}, i = 1, \ldots, K\}$, consisting of $K$ trajectories collected by the behavior policy $\mu$ in a delay-free environment (e.g., a simulator or other idealized demonstration system). At deployment, however, the policy operates under delayed feedback, where observations and rewards arrive after either a fixed latency of $\Delta$ steps, or a variable latency bounded by $\Delta$, which we treat as an effective fixed delay–a common simplification in control settings. In offline learning under this delayed MDP formulation, the agent must construct augmented state–action pairs from the delay-free dataset $\mathcal{D}$, enabling training in the delayed setting without new environment interaction. However, standard offline RL methods suffer from *out-of-distribution* (OOD) generalization issues when the learned policy queries actions not well-covered in $\mathcal{D}$ (Levine et al., 2020). To address this, policy-constrained approaches such as BRAC (Wu et al., 2019) and ReBRAC (Tarasov et al., 2023a) enforce similarity between the learned policy and the dataset's behavior policy, often by bounding a divergence measure $D(\pi, \mu)$. This motivates the policy-constrained learning objective under the delayed MDP:

$$\hat{Q}_{\Delta}^{\pi_{\Delta}} \leftarrow \arg\min_{Q_{\Delta}} \mathbb{E}_{(x,a,x')\sim\mathcal{D}}\left[\left(Q_{\Delta}(x,a) - \left(r_{\Delta}(x,a) + \gamma\mathbb{E}_{a'\sim\pi_{\Delta}(\cdot|x')}Q_{\Delta}(x',a')\right)\right)^2\right] \quad (1)$$

$$\pi_{\Delta}^{k+1} \leftarrow \arg\max_{\pi_{\Delta}} \mathbb{E}_{x\sim\mathcal{D}}\left[\mathbb{E}_{a\sim\pi_{\Delta}(\cdot|x)}[\hat{Q}_{\Delta}^{\pi_{\Delta}}(x,a)]\right] \quad \text{s.t.} \quad D(\pi_{\Delta}, \mu_{\Delta}) \leq \epsilon \quad (2)$$

Here, $Q_{\Delta}$ is the action-value function defined over augmented states, and $\mu_{\Delta}$ is the dataset's behavior policy lifted to the augmented space. The constraint margin $\epsilon$ controls the allowable divergence between the learned policy $\pi_{\Delta}$ and $\mu_{\Delta}$, mitigating OOD queries in the offline setting.

## 4 OUR APPROACH

We now present the **DT-CORL**, a novel offline RL framework for adapting online delayed feedback. DT-CORL integrates transformer-based belief modeling with policy-constrained offline learning to address the challenges outlined in previous sections. This section describes how we construct delay-compensated belief representations, train a policy using belief prediction, and incorporate policy regularization to ensure effective deployment under delay. Specifically, we introduce a belief-based policy-iteration framework that infers latent, delay-compensated states through a belief function, providing a compact and semantically grounded alternative. Then, we provide detailed algorithmic implementations. We further support this approach with a learning efficiency discussion to explain its efficiency benefits compared with the augmented approach.

### 4.1 BELIEF-BASED POLICY ITERATION

While the augmented-state formulation restores the Markov property in delayed MDPs, applying policy iteration in this space introduces key drawbacks, particularly under the offline setting. First, the effective state dimension grows from $|\mathcal{S}|$ to $|\mathcal{S}||\mathcal{A}|^{\Delta}$, increasing sample complexity and demanding significantly more data for reliable learning. Second, augmented trajectories reconstructed from delay-free datasets may not reflect the true delayed trajectories possibly happened online, leading to distribution mismatch and unstable value estimates. Lastly, treating action-history sequences as part of distinct augmented state inputs potentially discards some possible temporal order info, reducing sample efficiency and increasing overfitting risk. Therefore, trying to mitigate above problems, we propose our belief-based policy iteration framework, which essentially cast original augmented state space problem back into original state space via belief estimation to preserve temporal alignment

introduce by potential online delay. Following the BRAC framework (Wu et al., 2019), we first convert previously-defined constrained optimization to the unconstrained formulation for ease of optimization, where $\alpha_1$ and $\alpha_2$ are regularized constant.

$$\hat{Q}_\Delta^{\pi_\Delta} \leftarrow \arg\min_{Q_\Delta} \mathbb{E}_{(x,a,x')\sim\mathcal{D}} \left[ (Q_\Delta^{\pi_\Delta}(x,a) - (r_\Delta(x,a) + \gamma\mathbb{E}_{a'\sim\pi_\Delta^k(\cdot|x)}[Q_\Delta^{\pi_\Delta}(x',a')] - \alpha_1 \cdot D(\pi_\Delta^k, \mu_\Delta)))^2 \right]$$

$$\pi_\Delta^{k+1} \leftarrow \arg\max_{\pi_\Delta} \mathbb{E}_{x\sim\mathcal{D}} \left[ \mathbb{E}_{a\sim\pi_\Delta(\cdot|x)} \left[ \hat{Q}_\Delta^{\pi_\Delta}(x,a) \right] - \alpha_2 \cdot D(\pi_\Delta, \mu_\Delta) \right]$$

Moving forward, we need to map the delayed policy $\pi_\Delta$ and its Q function $\hat{Q}_\Delta^\pi$ back to the delay-free counterparts $\pi$ and $Q^\pi$ via the belief distribution $b_\Delta(s\,|\,x)$ introduced in Sec. 3. This requires (i) quantifying the performance gap between $\pi_\Delta$ and its belief-induced policy $\pi$, and (ii) relating the augmented value $\hat{Q}_\Delta^\pi(x,a)$ to the $Q^\pi(\hat{s},a)$ with $\hat{s} \sim b_\Delta(\cdot|x)$. Noted, throughout the remainder of this section we measure these discrepancies with the 1-Wasserstein distance (Villani et al., 2008).

**Lemma 4.1** (Delayed Performance Difference Bound (Wu et al., 2024b)). *For policies $\pi_{\Delta^\tau}$ and $\pi_\Delta$, with $\Delta^\tau < \Delta$. Given any $x \in \mathcal{X}$, if $Q_{\Delta^\tau}$ is $L_Q$-LC, the performance difference between policies can be bounded as follows:*

$$\mathbb{E}_{\substack{\hat{x}^\tau \sim b_\Delta(\cdot|x) \\ a\sim\pi_\Delta(\cdot|x)}} \left[ V_{\Delta^\tau}(\hat{x}^\tau) - Q_{\Delta^\tau}(\hat{x}^\tau, a) \right] \leq L_Q \mathbb{E}_{\hat{x}^\tau \sim b_\Delta(\cdot|x)} \left[ \mathcal{W}_1(\pi_{\Delta^\tau}(\cdot|\hat{x}^\tau)||\pi_\Delta(\cdot|x)) \right]$$

**Lemma 4.2** (Delayed Q-value Difference Bound(Wu et al., 2024b)). *For policies $\pi_{\Delta^\tau}$ and $\pi_\Delta$, with $\Delta^\tau < \Delta$. Given any $x \in \mathcal{X}$, if $Q_{\Delta^\tau}$ is $L_Q$-LC, the corresponding Q-value difference can be bounded as follows:*

$$\mathbb{E}_{\substack{a\sim\pi_\Delta(\cdot|x) \\ \hat{x}^\tau \sim b_\Delta(\cdot|x)}} \left[ Q_{\Delta^\tau}(\hat{x}^\tau, a) - Q_\Delta(x,a) \right] \leq \frac{\gamma L_Q}{1-\gamma} \mathbb{E}_{\substack{\hat{x}^\tau \sim b_\Delta(\cdot|x) \\ x'\sim\mathcal{P}_\Delta(\cdot|x,a) \\ a\sim\pi_\Delta(\cdot|x)}} \left[ \mathcal{W}_1(\pi_{\Delta^\tau}(\cdot|\hat{x}^\tau)||\pi_\Delta(\cdot|x)) \right]$$

Above two lemmas provide the exact quantification required. Moving along, we can convert $\pi_\Delta$ and $Q_\Delta^{\pi_\Delta}$ back to $\pi$ and $Q^\pi$. And, we can arrive the new policy-iteration framework in the original state space bridging by the belief function $b_\Delta$ as follows, where $\lambda_1$ and $\lambda_2$ are constants and $\mu_\Delta$ is the behavior policy after data augmentation. Specific derivation from augmented offline PI to belief-based PI in Eq. (3) and Eq. (4) can be found in App. B.1.

$$\hat{Q}^\pi \leftarrow \arg\min_Q \mathbb{E}_{(x,a,x')\sim\mathcal{D}} \Big[ \Big( \mathbb{E}_{\hat{s}\sim b_\Delta(\cdot|x)}[Q^\pi(\hat{s},a)] -$$
$$\Big( \mathbb{E}_{\hat{s}\sim b_\Delta(\cdot|x)}[r(\hat{s},a)] + \gamma \mathbb{E}_{\substack{\hat{s}'\sim b_\Delta(\cdot|x') \\ a'\sim\pi(\cdot|\hat{s}')}}[Q^\pi(\hat{s}',a')] - \lambda_1 \mathcal{W}_1(\pi, \mu_\Delta) \Big) \Big)^2 \Big]$$
$$(3)$$

$$\pi^{k+1} \leftarrow \arg\max_\pi \mathbb{E}_{x\sim\mathcal{D}} \left[ \mathbb{E}_{\substack{\hat{s}\sim b_\Delta(\cdot|x) \\ a\sim\pi(\cdot|\hat{s})}} \left[ \hat{Q}^\pi(\hat{s},a) \right] - \lambda_2 \mathcal{W}_1(\pi, \mu_\Delta) \right] \qquad (4)$$

**Proposition 4.3.** *Let the policy before and after the update in Eq. (4) be $\pi_{old}$ and $\pi_{new}$. Then after each policy evaluation in Eq. (3), we have $\mathbb{E}_{\bar{a}\sim\pi_{new}}[Q^{\pi_{new}}(s,\bar{a})] \geq \mathbb{E}_{\hat{a}\sim\pi_{old}}[Q^{\pi_{old}}(s,\hat{a})]$.*

Above proposition proves that iteratively applying Eq. (3) and Eq. (4) will monotonically increase Q value after update. Detailed proof can be found in App. B.2.

*Remark* 4.4. For deterministic MDP, $b_\Delta$ is also deterministic, meaning that $\mathbb{E}_{\hat{s}\sim b_\Delta(\cdot|x)}[r(\hat{s},a)] = r_\Delta(x,a)$ and $\mathbb{E}_{\hat{s}\sim b_\Delta(\cdot|x)}[Q^\pi(\hat{s},a)] = Q_\Delta^{\pi_\Delta}(x,a)$. Since $b_\Delta$ becomes an injection mapping under this setting, $\mathbb{E}_{\hat{s}\sim b_\Delta(\cdot|x)}[r(\hat{s},a)] = r(s,a)$ should also hold. Thus, we can directly leverage existing offline delay-free data tuples for update, except for the policy related terms.

A seemingly simpler alternative is a *two–stage* pipeline: (i) train a belief model offline; (ii) freeze it and apply any delay-free offline-RL algorithm to the original delay-free samples, deploying the resulting policy with that fixed belief function. This strategy, however, suffers from several drawbacks that DT-CORL avoids. **(i) Biased value targets.** In DT-CORL, the Bellman target in Eq. (3) is computed *through* the belief $b_\Delta(\cdot|x)$, so the critic learns on exactly the latent states the policy will later encounter. By contrast, the two–stage method treats the filter as error-free; residual belief error becomes unmodelled noise, forcing the critic to fit a moving target and biasing the learned $Q$-values.

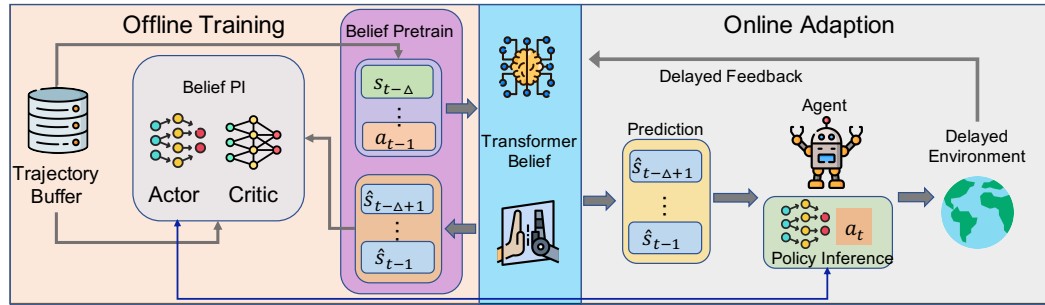

Figure 1: Overall pipeline of DT-CORL. In the Offline Training phase, trajectory data are augmented to train the transformer belief, and with the trained transformer belief, we conduct belief-based PI in the offline setting. In the Online Adaptation, we utilize the transformer belief to predict the current state from delayed observation, and adapt with offline-trained policy.

**(ii) Distribution mismatch.** Because the critic of the two-stage baseline is trained under $(s, a) \sim \mathcal{D}$ while the policy, evaluated through the frozen belief, quickly drifts to new action choices, the training and deployment distributions diverge. This mismatch is magnified in delayed environments and leads to sub-optimal returns, a phenomenon confirmed in our experiments (see Sec. 5). **(iii) Inferior sample efficiency in stochastic MDPs.** For stochastic dynamics, each offline tuple $(s, a, r, s')$ reveals only a single next state, whereas DT-CORL reuses the same transition to generate $\Delta$ latent states along the belief rollout, extracting richer supervision. Similar gains from "multiple synthetic samples per transition" have been observed in model-based offline RL (Kidambi et al., 2020; Yu et al., 2020). In short, embedding belief prediction *inside* the policy-iteration loop both contains model error and keeps the critic, policy, and belief model aligned—benefits a frozen two-stage pipeline cannot match, in addition to avoiding the dimensional explosion of raw state augmentation.

*Remark* 4.5. Existing work has compared the sample-complexity of augmentation- and belief-based methods in the *online* delayed-RL setting (Wu et al., 2024a;b). The same intuition should carry over to offline learning, but a formal analysis would require assumptions—e.g., uniform data-coverage, linear value realizability, and bounded rewards—that are standard in statistical offline-RL theory (Bai et al., 2022; Shi et al., 2022; Xie et al., 2021) yet do not hold in our setting. Establishing tight offline bounds without these restrictive conditions is therefore an open problem that we leave for future.

### 4.2 PRACTICAL IMPLEMENTATION

**Belief Prediction.** The belief prediction can be taken as a dynamic modeling problem. With a given offline trajectory $\{(s_t, a_t, r_t, s_{t+1})\}_{t=0}^H$, we can manually create the augmented state by stacking states and actions in $\Delta$ steps from start to end, where $x_t = \{s_{t-\Delta}, a_{t-\Delta}, a_{t-\Delta+1}, \cdots, a_{t-1}\}$ and the true state is $s_t$. Thus, the problem becomes training a belief function $b_\Delta$ which takes in $x_t$ and predict $s_t$. To reduce the potential compounding loss from belief prediction, we employ a transformer structure (Vaswani et al., 2017; Wu et al., 2025), where with the given augmented state $x_t$ transformer predicts a sequence from $\hat{s}_{t-\Delta+1}$ to $\hat{s}_t$. Based on the deterministic or stochastic nature of MDP, the transformer belief is updated with either MSE or MLE objectives. We validate the choice of transformer architecture in Sec. 5.3.

**Belief-based Policy Optimization.** Although many offline RL methods regularize the policy with KL, MMD, or Wasserstein distances (Levine et al., 2020; Wu et al., 2019), computing these divergences exactly is costly. Following the pragmatic approach of TD3 +BC and ReBRAC (Fujimoto & Gu, 2021; Tarasov et al., 2023a), we approximate the policy-behavior divergence by the mean-squared error between actions sampled from the learned policy and the augmented behavior policy. The policy-improvement step therefore, becomes

$$\pi^{k+1} = \arg\max_\pi \mathbb{E}_{(x,a)\sim\mathcal{D}}\Big[\mathbb{E}_{\substack{\hat{s}\sim b_\Delta(\cdot|x)\\ \hat{a}\sim\pi(\cdot|\hat{s})}}\big[\hat{Q}^{\pi^k}(\hat{s}, \hat{a})\big] - \alpha\big\|a - \hat{a}\big\|_2^2\Big]$$

where $\alpha > 0$ trades off exploitation and conservatism. This surrogate avoids training an explicit delay-augmented behavior model while retaining a simple quadratic penalty that is trivial to compute in continuous control. Besides, for deterministic MDPs the immediate reward $r(s_t, a_t)$ can be

Table 1: Normalized returns (%) on D4RL AntMaze tasks under deterministic and stochastic observation delays $\Delta \in \{4, 8, 16\}$. Results are averaged over 3 seeds. Best per column (including ties) is shown in **bold** with a light blue background.

| Setting | Method | umaze | | | umaze-diverse | | | medium-play | | | large-play | | |
|---|---|---|---|---|---|---|---|---|---|---|---|---|---|
| | | 4 | 8 | 16 | 4 | 8 | 16 | 4 | 8 | 16 | 4 | 8 | 16 |
| Deterministic | DBPT-SAC | 0.0 | 0.0 | 0.0 | 0.0 | 0.0 | 0.0 | 0.0 | 0.0 | 0.0 | **0.0** | 0.0 | 0.0 |
| | Augmented-BC | 60.7 | 62.3 | 31.0 | **69.0** | 58.7 | 30.0 | 0.0 | 1.67 | 1.0 | **0.0** | 0.3 | 0.0 |
| | Augmented-CQL | 72.3 | 44.0 | 12.7 | 27.7 | 23.7 | 22.0 | 0.33 | 1.67 | **4.67** | **0.0** | **1.33** | 0.67 |
| | Augmented-COMBO | 76.0 | 37.0 | 13.3 | 26.0 | 19.0 | 23.0 | **6.33** | **5.67** | 3.0 | **0.0** | **1.33** | **1.0** |
| | **DT-CORL** | **83.3** | **76.7** | **40.0** | 65.3 | **62.0** | **32.0** | 1.33 | 2.33 | 2.33 | **0.0** | 0.33 | 0.67 |
| Stochastic | DBPT-SAC | 0.0 | 0.0 | 0.0 | 0.0 | 0.0 | 0.0 | 0.0 | 0.0 | 0.0 | 0.0 | 0.0 | 0.0 |
| | Augmented-BC | 60.3 | 55.7 | 24.7 | 59.7 | 51.7 | 40.7 | 1.33 | 2.33 | 2.0 | 0.0 | 0.67 | 0.0 |
| | Augmented-CQL | 61.3 | 37.0 | 12.7 | 29.3 | 17.3 | 13.3 | 8.0 | **9.67** | **5.67** | **1.33** | 1.0 | 0.33 |
| | Augmented-COMBO | 64.7 | 36.7 | 13.3 | 32.7 | 15.0 | 12.7 | **8.67** | 7.67 | 3.67 | 0.67 | 1.33 | 1.0 |
| | **DT-CORL** | **88.3** | **88.0** | **67.3** | **74.0** | **58.3** | **56.0** | 2.00 | 2.67 | 3.33 | 0.0 | **1.67** | **1.67** |

read directly from the dataset, so no separate reward model is required (cf. Eq. (3), Eq. (4), and Remark 4.4).

**Online Adaptation.** At deployment time, we maintain a circular *action buffer* of length $\Delta$, initialized with random actions drawn from $\mathcal{A}$. At timestep $t$ the agent receives the delayed observation $o_t$ from the plant, appends the most recent action $a_{t-1}$ to the buffer, and forms the augmented input $x_t = \{o_t, a_{t-\Delta}, \dots, a_{t-1}\}$. This sequence is passed to the trained belief transformer, which returns a delay-compensated state estimate $\hat{s}_t$. The policy $\pi(\cdot \mid \hat{s}_t)$ then selects the next action $a_t$. During the first $\Delta$ steps (and symmetrically near episode termination), the buffer is not yet full. We handle these boundary conditions by inserting a special [MASK] token for missing actions and enabling the transformer's built-in masking mechanism, ensuring consistent state prediction throughout the episode. Detailed neural network structures and hyper-parameters can be found in App. D.

## 5 EXPERIMENTS

To evaluate the effectiveness of our approach, we conduct experiments on the standard D4RL benchmark suite, including both MuJoCo locomotion and AntMaze goal-conditioned tasks. Our experimental analysis highlights two primary advantages of DT-CORL. First, we demonstrate that our method significantly outperforms both traditional offline RL algorithms with state augmentation adaptation and the SOTA Online delay-robust RL algorithm. Second, we show that DT-CORL surpasses hybrid approaches that combine separately trained delay-belief models with existing offline RL algorithms, underscoring the benefit of belief-involved policy evaluation. In our ablation studies, we examine how trajectory availability impacts performance and compare alternative belief architectures, including ensemble MLPs and diffusion-based predictors, to validate the choice of the transformer-based belief model.

### 5.1 ANTMAZE

We benchmark **DT-CORL** on the AntMaze goal-conditioned tasks from the D4RL offline RL suite (Fu et al., 2020; Todorov et al., 2012). Since no existing method is designed to learn a delay-robust policy solely from delay-free offline data, we construct two baseline methods for comparison: (i) *Augmented-BC*, which applies standard behavioral cloning (Torabi et al., 2018) in a $\Delta$-step augmented state space; and (ii) *Augmented-CQL*, which runs Conservative Q-Learning (Kumar et al., 2020) on the same augmented state representation. Additionally, we compare DT-CORL against a model-based offline RL baseline, *Augmented-COMBO* (Yu et al., 2021), and the state-of-the-art delay-robust RL algorithm *DBPT-SAC* (Wu et al., 2025). We evaluate all methods on four standard AntMaze environments: medium-play, large-play, umaze-diverse, and umaze. For both **deterministic** and **stochastic delay** settings, we test on 4, 8, and 16 steps of delay respectively. For stochastic delay, it means that the delay at each step follows a uniform distribution, $\Delta \in \mathcal{U}(1, k), k \in \{4, 8, 16\}$. From Table 1, online delayed RL methods such as DBPT-SAC collapse under the offline setting, confirming online methods' incompatibility with offline setting. In umaze and umaze-diverse, DT-CORL consistently outperforms all baselines and shows far less degradation as delay length increases, whereas augmentation-based methods deteriorate sharply. On the harder medium-play

Table 2: DT-CORL vs. belief-based baselines (Belief-CQL, Belief-IQL) on D4RL MuJoCo tasks. Normalized returns (%). Delays: deterministic $\Delta \in \{4, 8, 16\}$, stochastic $\Delta \sim \mathcal{U}(1, k)$, $k \in \{4, 8, 16\}$. Best results per column are shown in **bold** with light blue background.

| Task | Method | medium | | | med-expert | | | med-replay | | | expert | | |
|---|---|---|---|---|---|---|---|---|---|---|---|---|---|
| | | 4 | 8 | 16 | 4 | 8 | 16 | 4 | 8 | 16 | 4 | 8 | 16 |
| **Deterministic Delays** | | | | | | | | | | | | | |
| **Hopper** | IQL | 5.3 | 5.8 | 4.8 | 5.5 | 6.4 | 5.3 | 5.8 | 6.4 | 2.4 | 4.5 | 9.0 | 5.6 |
| | CQL | 8.2 | 4.3 | 4.3 | 7.3 | 3.4 | 5.9 | 8.6 | 4.4 | 6.1 | 7.9 | 3.9 | 4.1 |
| | Belief-IQL | 27.7 | 29.3 | 25.4 | 26.7 | 26.6 | 24.7 | 24.7 | 25.5 | 23.4 | 18.7 | 17.4 | 15.9 |
| | Belief-CQL | 75.4 | 56.8 | 42.9 | 92.9 | 39.5 | 35.2 | **110.1** | 99.7 | 96.6 | 81.1 | 43.3 | 45.9 |
| | DT-CORL | **79.4** | **85.0** | **71.8** | **113.0** | **112.2** | **109.9** | 99.4 | **100.8** | **100.2** | **112.9** | **113.1** | **112.2** |
| **HalfCheetah** | Belief-IQL | 30.8 | 10.6 | 5.3 | 24.8 | 6.1 | 3.3 | 23.3 | 13.8 | 9.7 | 6.8 | 4.8 | 3.6 |
| | Belief-CQL | **49.2** | 8.9 | 3.0 | 22.7 | 6.5 | 1.5 | 36.1 | 14.4 | 6.4 | 1.5 | 1.5 | 1.5 |
| | DT-CORL | 47.4 | **27.8** | **6.4** | **44.7** | **21.3** | **8.7** | **43.6** | **27.1** | **7.9** | **20.6** | **5.1** | **5.2** |
| **Walker2d** | Belief-IQL | 33.4 | 25.7 | 24.6 | 49.6 | 17.3 | 16.4 | 28.2 | 20.6 | 18.3 | 25.5 | 19.9 | 16.7 |
| | Belief-CQL | 87.0 | 64.1 | 39.2 | 105.8 | 99.5 | 51.0 | 93.3 | **93.5** | 61.0 | **111.1** | 110.8 | 97.7 |
| | DT-CORL | **87.4** | **87.6** | **86.8** | **112.1** | **112.0** | **118.1** | **93.6** | 90.5 | **88.1** | 110.9 | **111.2** | **110.5** |
| **Stochastic Delays** | | | | | | | | | | | | | |
| **Hopper** | IQL | 8.3 | 6.0 | 12.9 | 9.4 | 6.4 | 10.4 | 8.6 | 4.0 | 2.9 | 11.4 | 3.1 | 11.6 |
| | CQL | 8.9 | 4.3 | 11.1 | 7.4 | 6.0 | 11.5 | 6.1 | 5.3 | 9.1 | 6.9 | 3.4 | 4.2 |
| | Belief-IQL | 27.4 | 27.4 | 28.4 | 28.3 | 27.9 | 23.8 | 25.0 | 24.5 | 26.0 | 18.6 | 16.7 | 16.9 |
| | Belief-CQL | **80.8** | **67.8** | 74.3 | 91.8 | 48.9 | 57.5 | **100.8** | 99.8 | **99.2** | 72.3 | 35.4 | 20.1 |
| | DT-CORL | 78.5 | 72.1 | **79.3** | **113.6** | **112.7** | **85.4** | 99.4 | **100.1** | 98.8 | **113.2** | **112.8** | **113.1** |
| **HalfCheetah** | Belief-IQL | 33.4 | 31.2 | 21.6 | 31.1 | 16.3 | 8.2 | 24.9 | 20.0 | 15.3 | 13.1 | 9.2 | 4.3 |
| | Belief-CQL | **52.6** | 46.6 | 16.2 | 48.0 | 22.1 | 3.6 | **47.1** | 41.4 | 19.7 | 6.5 | 2.8 | 1.7 |
| | DT-CORL | 48.2 | **47.5** | **38.4** | **70.0** | **44.3** | **31.7** | 47.7 | **43.3** | **30.4** | **85.1** | **12.7** | **5.8** |
| **Walker2d** | Belief-IQL | 34.6 | 37.5 | 36.7 | 49.4 | 25.0 | 20.4 | 31.4 | 25.4 | 24.2 | 43.1 | 27.7 | 20.7 |
| | Belief-CQL | 84.8 | 81.7 | 78.9 | 112.1 | 106.5 | 85.1 | **94.9** | **97.7** | **95.3** | **111.1** | **111.1** | 109.6 |
| | DT-CORL | **86.8** | **87.4** | **87.0** | **114.1** | **113.6** | **111.5** | 93.0 | 90.9 | 91.8 | 110.9 | 110.9 | **110.5** |

and `large-play` tasks, overall performance for all methods remains low; the poor results across methods suggest that additional goal-conditioned modifications may be required. Overall, DT-CORL demonstrates stronger robustness to increasing delays than augmentation-based baselines. We also observe that augmentation-based methods degrade more under stochastic delays, as the additional randomness introduces variance that destabilizes policy learning. In contrast, our belief-based approaches exhibit the opposite trend: the reduced sample complexity and improved temporal prediction allow them to benefit from the effectively shorter delays under stochastic settings. We further justify our method's superiority under various environments in additional MuJoCo tasks. Detailed results can be found in Table 7 and Table 8.

## 5.2 BELIEF-BASED COMPARISON

To justify the effectiveness of incorporating belief estimation in offline policy optimization, we compare our method DT-CORL with the following baselines: (i) *Offline RL*, naive implementations of CQL and IQL without any delayed adaption. (ii) *Belief-CQL*, which feeds the CQL algorithm the transformer belief used by DT-CORL instead of raw augmentation. (iii) *Belief-IQL*, which runs Implicit Q-Learning Kostrikov et al. (2021) with the same delayed belief above. We test all methods on D4RL MuJoCo suite following the `medium`, `medium-expert`, `medium-replay`, and `expert` trajectory setting. For both **deterministic** and **stochastic** delay, we adopt the setting described in the previous subsection. From Table 2, we can tell the clear performance gap between other naive belief-based methods and DT-CORL, and the ineffectiveness of naive CQL and IQL under our delayed setting. Belief-IQL relies heavily on implicit Q-learning updates, which are sensitive to inaccuracies in the latent belief state. Specifically, IQL's actor update weights actions by $\omega = \exp(A/\lambda)$, and under delayed/partial observations the same belief state carries noisy $Q - V$; tiny errors explode after the exponential, producing unstable, off-support policy updates. Thus, without alignment between belief learning and policy training, small belief prediction errors can lead to large value overestimation or underestimation, which end up with poor performance of Belief-IQL across all the scenarios. In Belief-CQL, the Q-function is trained on the precomputed belief embeddings. This decoupling leads to suboptimal value estimates, especially under long delays

Figure 2: Step-by-step detailed comparison of prediction accuracy for different models.

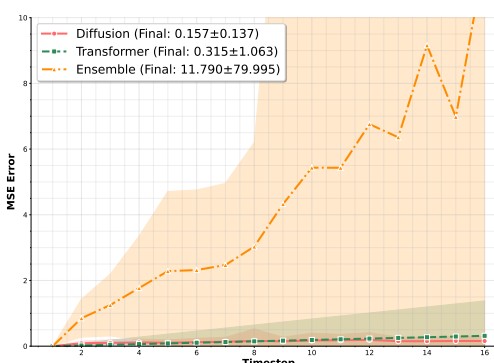

| Model | Total Parameters (M) | Inference (ms) |
|---|---|---|
| Ensemble MLP | 1.80 | 69.50±2.86 |
| Transformer | 7.87 | 22.27±9.02 |
| Diffusion | 3.96 | 665.21±55.5 |

Table 3: Comparison of total parameters (M) and inference speed (ms) for each model up to 16 steps.

| Data | DT-CORL | Belief-CQL | Aug-CQL |
|---|---|---|---|
| 25% | 9.8 | 3.08 | 2.47 |
| 50% | 12.0 | 3.80 | 2.80 |
| 75% | 15.3 | 4.05 | 3.89 |
| 100% | 27.8 | 8.90 | 3.80 |

Table 4: Performance of DT-CORL under different data availability levels for the `HalfCheetah-medium-v2`.

where belief inaccuracies compound. This tendency can be spotted across various tasks with different trajectory settings. DT-CORL aligns the belief prediction with downstream policy evaluation and optimization, allowing both the transformer and Q-function to co-adapt under delayed feedback. Besides, by conditioning the policy and critic on belief-derived latent states, DT-CORL mitigates the delay-induced distribution shift more effectively than purely augmenting state spaces or pretraining belief models.

### 5.3 ABLATION STUDIES

**Why Transformer?**    In this subsection, we justify our choice of a transformer-based belief model by comparing it against two common alternatives—ensemble MLPs and diffusion models. We evaluate all models in the Hopper-medium setting, measuring (i) multi-step prediction accuracy (MSE up to 16 steps; Figure 2) and (ii) inference speed and parameter count (Table 3). All models are trained for 100 epochs on the same `medium` dataset and evaluated in the environment; implementation details appear in App. D. Results (See Figure 2 and Table 3) highlight a clear trade-off: ensemble MLPs are lightweight but accumulate error rapidly over long horizons, degrading belief quality; diffusion models achieve strong accuracy but incur high computational cost due to iterative denoising. In contrast, the transformer model provides the best balance, offering strong multi-step accuracy, stable long-horizon predictions, and fast inference suitable for online deployment. Additional qualitative prediction visualizations are included in App. C.1. In addition, we further analyze the impact of transformer size on sample efficiency and prediction accuracy. Detailed experiments can also be found in App. C.1 and Table 9.

**Trajectories Availability.**    We next examine how offline data availability influences the performance of DT-CORL and its baselines. Using the `HalfCheetah-medium-v2` environment with a fixed delay of 8 steps, we vary the proportion of the offline dataset available for training: 25As shown in Table 4, all methods improve with more data, but DT-CORL consistently achieves the highest returns across all data levels. Notably, the performance gap between DT-CORL and the belief-based or augmentation-based baselines widens as more data becomes available, while those baselines show only mild gains. These results indicate that DT-CORL not only maintains superior performance in low-data regimes but also scales more effectively with additional data—highlighting the benefit of jointly learning the belief model and policy within a unified framework.

**Joint Training Benefits.**    To empirically validate this coupling, we compare DT-CORL against a variant that uses a separately pretrained belief model with no further adaptation during policy learning. As shown in Table 5, joint training consistently outperforms separate pretraining across all delay settings in the Hopper suite—yielding significantly higher returns at both small and large delays. This result confirms that joint optimization reduces offline-to-online distribution mismatch and enables the belief model to specialize to the value function's error landscape, providing substantially more stable delayed-policy learning.

**Delay Robustness.**    To assess DT-CORL's reliability under challenging latency conditions, we evaluate two robustness settings: (i) scaling to long-horizon delays (32–64 steps), and (ii) testing

Table 5: Joint vs. Separate Belief Training

| Training Mode | 4 | 8 | 16 |
|---|---|---|---|
| Separate | 92.1 / 91.7 | 81.4 / 87.4 | 68.3 / 73.1 |
| DT-CORL | **101.2 / 101.2** | **102.8 / 99.4** | **98.5 / 94.2** |

Table 6: DT-CORL Across Delay Distributions

| Method | Uniform | Gauss | Exp | Binom |
|---|---|---|---|---|
| DT-CORL | 79.3 | 82.1 | 85.8 | 77.4 |

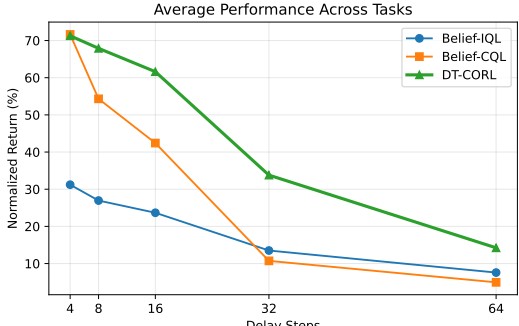

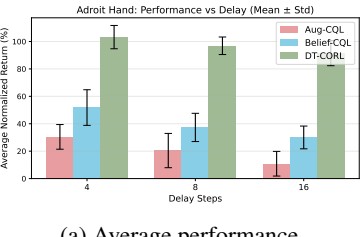
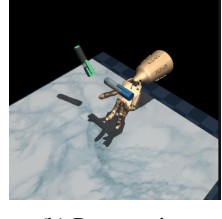
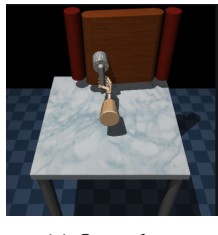
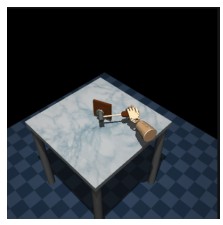

(a) Average performance     (b) Pen rotation     (c) Open door     (d) Hammer smash

Figure 3: (a) Describes the average performance of *Aug-CQL*, *Belief-CQL*, and *DT-CORL* across three dexterous hand manipulation tasks (a)-(c) under various delay setting ranging from 4 to 16.

under alternative delay distributions that share the same expected delay but differ in variance and temporal structure. All distributional experiments use `Hopper-medium-v2`, with maximum delay fixed at 16 and Gaussian, exponential, binomial, and uniform delays parameterized to have mean 8 for fair comparison. Top right figure shows average normalized return across the three MuJoCo locomotion tasks as delays increase, averaged over deterministic and stochastic variants. As expected, performance decreases with larger delays, highlighting the increased difficulty; full results appear in Table 10 and App. C.1. From Table 6, we also observe that DT-CORL performs robustly across different delay distributions—even without prior knowledge of the true online delay process—demonstrating strong generalization of our belief estimation module.

## 5.4 DEXTEROUS MANIPULATION

To evaluate delay robustness in high-dimensional, contact-rich manipulation, we benchmark all methods on the Dexterous Hand Manipulation suite (Zhu et al., 2019), using the expert demonstrations provided and applying standard delay setting as before. These tasks involve discontinuous contacts, multi-finger coordination, and highly sensitive state-action coupling, making delayed observation particularly challenging. As shown in Fig. 3, DT-CORL consistently attains the highest performance across all tested delay levels and exhibits substantially more graceful degradation compared to *Aug-CQL* and *Belief-CQL*. This indicates that our belief model captures fine-grained temporal structure that is critical for manipulation under latency. Full per-task results for Pen, Door, and Hammer appear in Table 11, further highlighting DT-CORL's robustness, particularly under for the Hammer task where both augmented and belief-based baselines failed.

## 6 CONCLUSION

We presented **DT-CORL**, the first framework for offline-to-online delay adaptation that learns delay-robust policies *solely* from delay-free data. By combining a transformer-based belief predictor with conservative, behavior-regularized policy iteration, DT-CORL avoids the dimensional blow-up of history augmentation while mitigating out-of-distribution errors common in offline RL. Experiments on various tasks show consistent gains over both augmented-state and belief-based baselines, under deterministic and stochastic delays. Looking forward, several extensions remain open. First, DT-CORL currently assumes known, fixed delays; extending it to handle *unknown*, *time-varying*, or *heterogeneous* delays is an important next step. Notably, such cases remain largely unaddressed in both online and offline delayed RL. Second, scaling the approach to high-dimensional perceptual domains such as vision-based manipulation may require spatial attention or structured world models.

## ACKNOWLEDGMENT

Simon Sinong Zhan, Xiangyu Shi, and Qi Zhu acknowledge the support from National Science Foundation grants 2324936, 2328973, 2328032, and 2133630. Qingyuan Wu and Chao Huang are supported by the grant EP/Y002644/1 under the EPSRC ECR International Collaboration Grants program, funded by the International Science Partnerships Fund (ISPF) and the UK Research and Innovation.

## ETHICS STATEMENT

We affirm that all authors have read and adhere to the ICLR Code of Ethics. Our work does not involve human or animal subjects, sensitive personal data, or privacy risks. The experiments are fully based on public benchmarks (D4RL), with no proprietary or private datasets, ensuring transparency and reproducibility. We used Large Language Models only for manuscript writing, formatting, and routine data-processing; all algorithmic design, theoretical derivations, and experimental evaluations are solely the work of the authors. There are no known immediate risks of misuse from our method; however, we recognize that deployment in safety-critical systems under delays might require careful calibration.

## REPRODUCIBLE STATEMENT

To ensure reproducibility of all experimental results, we provide the following supporting materials and practices. The code implementing DT-CORL, including training scripts, belief model architectures, and evaluation pipelines, has been made available in an the following repo (`https://github.com/SimonZhan-code/DT-CORL`). All algorithmic assumptions, hyperparameters (learning rates, network architectures, regularization weights, transformer depth, etc.), and training settings are described in detail in Sec. 5 of the main paper, and additional implementation details are given in App. D. The datasets used are standard benchmarks (D4RL locomotion and AntMaze tasks). We report performance averaged over multiple random seeds. All experiment results are listed in Sec. 5 and App. C.

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

## A  LLM Usage Statement

The usage of LLMs in this work is limited to paper writing support, language refinement, and experimental data processing. Specifically, LLMs assisted in improving the clarity and coherence of the manuscript, generating LaTeX tables and formatting results for presentation. Importantly, LLMs were not involved in the design of algorithms, the development of theoretical results, or the execution of experiments, ensuring that all core scientific contributions remain entirely the work of the authors.

## B  Belief-based Policy Iteration

### B.1  Policy Iteration

To arrive the final derivation, we need the following lemma.

**Lemma B.1.** *(General Delayed Performance Difference (Wu et al., 2024b))  For policies $\pi$ and $\mu_\Delta$ with any $x \in \mathcal{X}$, the performance difference is denoted as $I(x)$*

$$I(x) = \mathbb{E}_{\hat{s}\sim b(\cdot|x)}[V(s)] - V_{\Delta,\beta}(x)$$

$$= \frac{1}{1-\gamma} \mathbb{E}_{\substack{\hat{s}\sim b(\cdot|x)\\a\sim\mu_\Delta(\cdot|x)\\x\sim\mathcal{D}}} [V(\hat{s}) - Q(\hat{s},a)]$$

For next step, we try to cast the above optimization problem back to the ordinary state/action space with belief function $b$ defined above. To begin with, we can convert $Q_\Delta(x,a)$ back to $\mathbb{E}_{\hat{s}\sim b(\cdot|x)}[Q(\hat{s},a)]$ using Lem. 4.1.

$$\mathbb{E}_{\substack{(x,a,x')\sim\mathcal{D}\\\hat{s}\sim b(\cdot|x)\\\hat{s}'\sim b(\cdot|x')}} [Q(\hat{s},a) - Q_\Delta(x,a)]$$

$$= \mathbb{E}_{\substack{(x,a,x')\sim\mathcal{D}\\\hat{s}\sim b(\cdot|x)}} \left[r(\hat{s},a) + \gamma\mathbb{E}_{\hat{s}'\sim b(\cdot|x')}[V(\hat{s}')]\right] - \mathbb{E}_{(x,a,x')\sim\mathcal{D}}[r(x,a) + V_\Delta(x')]$$

$$= \gamma\mathbb{E}_{\substack{(x,a,x')\sim\mathcal{D}\\\hat{s}\sim b(\cdot|x)}} \left[\mathbb{E}_{\hat{s}'\sim b(\cdot|x')}[V(\hat{s}')] - V(x')\right]$$

$$\leq \frac{\gamma L_Q}{1-\gamma} \mathbb{E}_{\substack{(x,a,x')\sim\mathcal{D}\\\hat{s}\sim b(\cdot|x)}} [\mathcal{W}_1(\pi(\cdot|\hat{s})||\mu_\Delta(\cdot|x))]$$

Similarly, we can extend it to other term in the policy evaluation part.

$$\mathbb{E}_{\substack{x'\sim\mathcal{D}\\\hat{s}'\sim b(\cdot|x')\\a'\sim\pi_\Delta(\cdot|x')}} [Q(\hat{s}',a') - Q_\Delta(x',a')]$$

$$= \mathbb{E}_{\substack{\hat{s}'\sim b(\cdot|x')\\a'\sim\pi_\Delta(\cdot|x')\\x'\sim\mathcal{D}}} [Q(\hat{s}',a')] - \mathbb{E}_{\substack{a'\sim\pi_\Delta(\cdot|x')\\x'\sim\mathcal{D}}} [Q_\Delta(x',a')]$$

$$\leq \frac{\gamma L_Q}{1-\gamma} \mathbb{E}_{\substack{x'\sim\mathcal{D}\\\hat{s}'\sim b(\cdot|x')}} [\mathcal{W}_1(\pi(\cdot|\hat{s}')||\pi_\Delta(\cdot|x'))]$$

Then, we can start to break down the policy evaluation defined above.

$$\mathbb{E}_{(x,a,x')\sim\mathcal{D}}\left[\left(\widehat{Q}_\Delta^\pi(x,a) - r_\Delta(x,a) - \gamma\mathbb{E}_{a'\sim\pi_\Delta^k(\cdot|x')}[\widehat{Q}_\Delta^\pi(x',a')] + \alpha_1 D(\pi_\Delta^k,\mu_\Delta)\right)^2\right]$$

$$\Leftrightarrow \mathbb{E}_{(x,a,x')\sim\mathcal{D}}\left[\left(\mathbb{E}_{\hat{s}\sim b(\cdot|x)}\left[Q^\pi(\hat{s},a) - \frac{\gamma L_Q}{1-\gamma}\mathcal{W}_1(\pi(\cdot|\hat{s}),\mu_\Delta(\cdot|x))\right]\right.\right.$$

$$\left.\left. - \left(\mathbb{E}_{\hat{s}\sim b(\cdot|x)}[r(\hat{s},a)] + \gamma\mathbb{E}_{\hat{s}'\sim b(\cdot|x')}\left[Q^\pi(\hat{s}',a') - \frac{\gamma L_Q}{1-\gamma}\mathcal{W}_1(\pi(\cdot|\hat{s}'),\pi_\Delta(\cdot|x'))\right] - \alpha_1\mathcal{W}_1(\pi_\Delta,\mu_\Delta)\right)\right)^2\right].$$

The next step is trying to sort out all the policy divergence term, and convert the expectation term with respect to $\pi_\Delta$ back to $\pi$. For simplification, let's define $c = \frac{\gamma L_Q}{1-\gamma}$ and hide the policy divergence term for now. We have the following:

$$\mathbb{E}_{(x,a,x')\sim\mathcal{D}}\left[\left(\mathbb{E}_{\hat{s}\sim b(\cdot|x)}\left[Q^\pi(\hat{s},a)\right] - \left(\mathbb{E}_{\hat{s}\sim b(\cdot|x)}\left[r(\hat{s},a)\right] + \gamma \mathbb{E}_{\substack{\hat{s}'\sim b(\cdot|x')\\ a'\sim\pi_\Delta(\cdot|x')}}\left[Q^\pi(\hat{s}',a')\right]\right)\right)^2\right]$$

According to Lipchitz continuous assumptions on Q function Liotet et al. (2022), we can derive that $\left|\mathbb{E}_{\substack{a_1\sim\mu\\ a_2\sim\nu}}\left[Q^\pi(s,a_1) - Q^\pi(s,a_2)\right]\right| \le L_Q\mathcal{W}_1(\mu,\nu) \quad \forall s \in \mathcal{S}$. Thus, we can derive the following:

$$\mathbb{E}_{(x,a,x')\sim\mathcal{D}}\left[\left(\mathbb{E}_{\hat{s}\sim b(\cdot|x)}\left[Q^\pi(\hat{s},a)\right] - \left(\mathbb{E}_{\hat{s}\sim b(\cdot|x)}\left[r(\hat{s},a)\right] + \gamma\mathbb{E}_{\substack{\hat{s}'\sim b(\cdot|x')\\ a'\sim\pi(\cdot|\hat{s}')}}\left[Q^\pi(\hat{s}',a')\right] + (1-\gamma)c\mathcal{W}_1(\pi,\pi_\Delta)\right)\right)^2\right]$$

$$(5)$$

Now, we have the policy evaluation in the original state space format. Let's take a close look at the remaining policy divergence terms.

$$\gamma c\mathcal{W}_1(\pi,\pi_\Delta) + \alpha_1\mathcal{W}_1(\pi_\Delta,\mu_\Delta) - c\mathcal{W}_1(\pi,\mu_\Delta) + (1-\gamma)c\mathcal{W}_1(\pi,\pi_\Delta)$$
$$= c\mathcal{W}_1(\pi,\pi_\Delta) + \alpha_1\mathcal{W}_1(\pi_\Delta,\mu_\Delta) - c\mathcal{W}_1(\pi,\mu_\Delta)$$

The triangle inequality holds for the Wasserstein distance. Specifically, $\mathcal{W}_p(\mu,\rho) \le \mathcal{W}_p(\mu,\nu) + \mathcal{W}_p(\nu,\rho)$ for all $\mu$, $\nu$, and $\rho$ in the same metric space and $p \ge 1$ (Villani et al., 2008). With the proper choice of $\alpha$, it is easy to unify all the 1-Wasserstein terms to $\pi$ and behavior policy. Besides, since all the 1-Wasserstein terms are bounded here, it won't affect the convergence property of policy iteration.

$$\hat{Q}^\pi \leftarrow \arg\min_Q \mathbb{E}_{(x,a,x')\sim\mathcal{D}}\left[\left(\mathbb{E}_{\hat{s}\sim b(\cdot|x)}\left[Q^\pi(\hat{s},a)\right]\right.\right.$$
$$\left.\left.\left(\mathbb{E}_{\hat{s}\sim b(\cdot|x)}\left[r(\hat{s},a)\right] + \gamma\mathbb{E}_{\substack{\hat{s}'\sim b(\cdot|x')\\ a'\sim\pi(\cdot|\hat{s}')}}\left[Q^\pi(\hat{s}',a')\right] - \alpha_1\mathcal{W}_1(\pi,\mu_\Delta)\right)\right)^2\right].$$

Using the derivation above, we can easily reformulate the policy improvement back to the original state space.

$$\mathbb{E}_{x\sim\mathcal{D}}\left[\mathbb{E}_{a\sim\pi_\Delta(\cdot|x)}\left[\hat{Q}_\Delta^{\pi_\Delta}(x,a)\right] - \alpha_2 \cdot D(\pi_\Delta,\mu_\Delta)\right]$$
$$\Leftrightarrow\mathbb{E}_{x\sim\mathcal{D}}\left[\mathbb{E}_{\substack{\hat{s}\sim b(\cdot|x)\\ a\sim\pi_\Delta(\cdot|x)}}\left[\hat{Q}^\pi(\hat{s},a)\right] - \frac{\gamma L_Q}{1-\gamma}\mathcal{W}_1(\pi,\pi_\Delta) - \alpha_2\mathcal{W}_1(\pi_\Delta,\mu_\Delta)\right]$$
$$\Leftrightarrow\mathbb{E}_{x\sim\mathcal{D}}\left[\mathbb{E}_{\substack{\hat{s}\sim b(\cdot|x)\\ a\sim\pi(\cdot|\hat{s})}}\left[\hat{Q}^\pi(\hat{s},a)\right] + (1-\gamma)c\mathcal{W}_1(\pi,\pi_\Delta) - c\mathcal{W}_1(\pi,\pi_\Delta) - \alpha_2\mathcal{W}_1(\pi_\Delta,\mu_\Delta)\right]$$
$$\Leftrightarrow\mathbb{E}_{x\sim\mathcal{D}}\left[\mathbb{E}_{\substack{\hat{s}\sim b(\cdot|x)\\ a\sim\pi(\cdot|\hat{s})}}\left[\hat{Q}^\pi(\hat{s},a)\right] - (\gamma c\mathcal{W}_1(\pi,\pi_\Delta) + \alpha_2\mathcal{W}_1(\pi_\Delta,\mu_\Delta))\right]$$

Using a similar trick mentioned above, with appropriate selection of $\alpha_2$, we can combine the above two policy divergence terms into one.

$$\mathbb{E}_{x\sim\mathcal{D}}\left[\mathbb{E}_{\substack{\hat{s}\sim b(\cdot|x)\\ a\sim\pi(\cdot|\hat{s})}}\left[\hat{Q}^\pi(\hat{s},a)\right] - \gamma c\mathcal{W}_1(\pi,\mu_\Delta)\right]$$

## B.2 POLICY IMPROVEMENT

*Proof.*

$$\pi_{new} = \arg\max_\pi \mathbb{E}_{x\sim\mathcal{D}}\left[\mathbb{E}_{\substack{\hat{s}\sim b(\cdot|x)\\ a\sim\pi(\cdot|\hat{s})}}\left[\hat{Q}^\pi(\hat{s},a)\right] - \alpha\mathcal{W}_1(\pi,\mu_\Delta)\right]$$

By following Eq. (4), we can easily have have

$$\mathbb{E}_{x \sim \mathcal{D}} \left[ \mathbb{E}_{\substack{\hat{s} \sim b(\cdot|x) \\ a \sim \pi_{new}(\cdot|\hat{s})}} \left[ \hat{Q}^\pi_{new}(\hat{s}, a) \right] - \alpha \mathcal{W}_1(\pi_{new}, \mu_\Delta) \right]$$

$$\geq \mathbb{E}_{x \sim \mathcal{D}} \left[ \mathbb{E}_{\substack{\hat{s} \sim b(\cdot|x) \\ a \sim \pi_{old}(\cdot|\hat{s})}} \left[ \hat{Q}^\pi_{old}(\hat{s}, a) \right] - \alpha \mathcal{W}_1(\pi_{old}, \mu_\Delta) \right]$$

$$\hat{Q}^\pi_{new}(\hat{s}, a') = \mathbb{E}_{(x,a,x') \sim \mathcal{D}} \left[ \mathbb{E}_{\hat{s} \sim b(\cdot|x)} \left[ r(\hat{s}, a) \right] + \gamma \mathbb{E}_{\substack{\hat{s}' \sim b(\cdot|x') \\ a' \sim \pi_{new}(\cdot|\hat{s}')}} \left[ Q^\pi_{new}(\hat{s}', a') \right] - \alpha \mathcal{W}_1(\pi_{new}, \mu_\Delta) \right]$$

$$\geq \mathbb{E}_{(x,a,x') \sim \mathcal{D}} \left[ \mathbb{E}_{\hat{s} \sim b(\cdot|x)} \left[ r(\hat{s}, a) \right] + \gamma \mathbb{E}_{\substack{\hat{s}' \sim b(\cdot|x') \\ a' \sim \pi_{old}(\cdot|\hat{s}')}} \left[ Q^\pi_{old}(\hat{s}', a') \right] - \alpha \mathcal{W}_1(\pi_{old}, \mu_\Delta) \right]$$

$$= \hat{Q}^\pi_{old}(\hat{s}, a')$$

$\square$

## C    EXPERIMENTS DETAILS

### C.1    ADDITIONAL RESULTS

In this part, we provide visualization of the delay belief prediction over a 16-step trajectory rollout in the HOPPER environment. Each frame corresponds to a simulation step under a delayed control setting. The agent executes actions conditioned on the predicted belief state rather than the true current observation. The sequence illustrates how the learned delayed belief model tracks and reconstructs the latent state dynamics despite observation–action delays. Accurate belief predictions enable the agent to maintain coherent behavior across all steps.

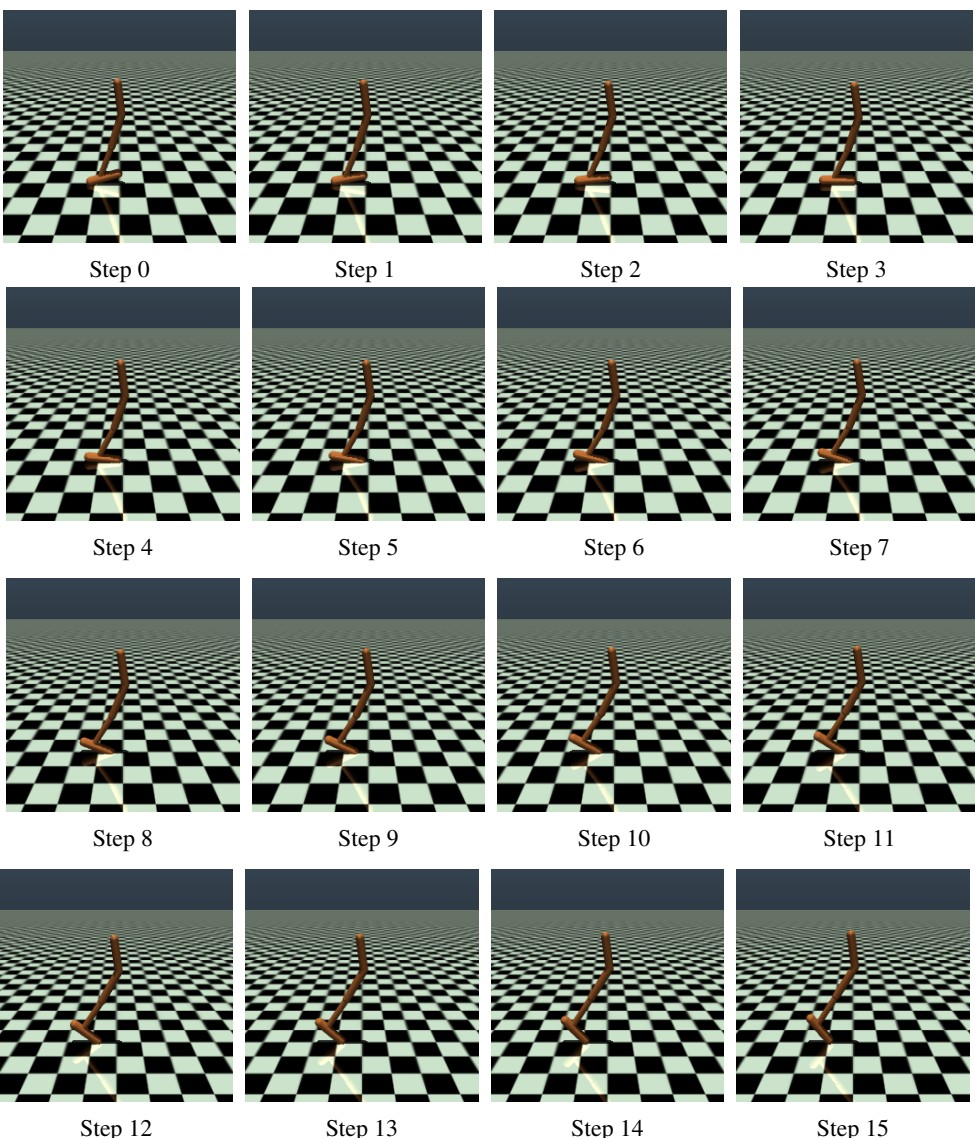

Figure 4: Ground Truth

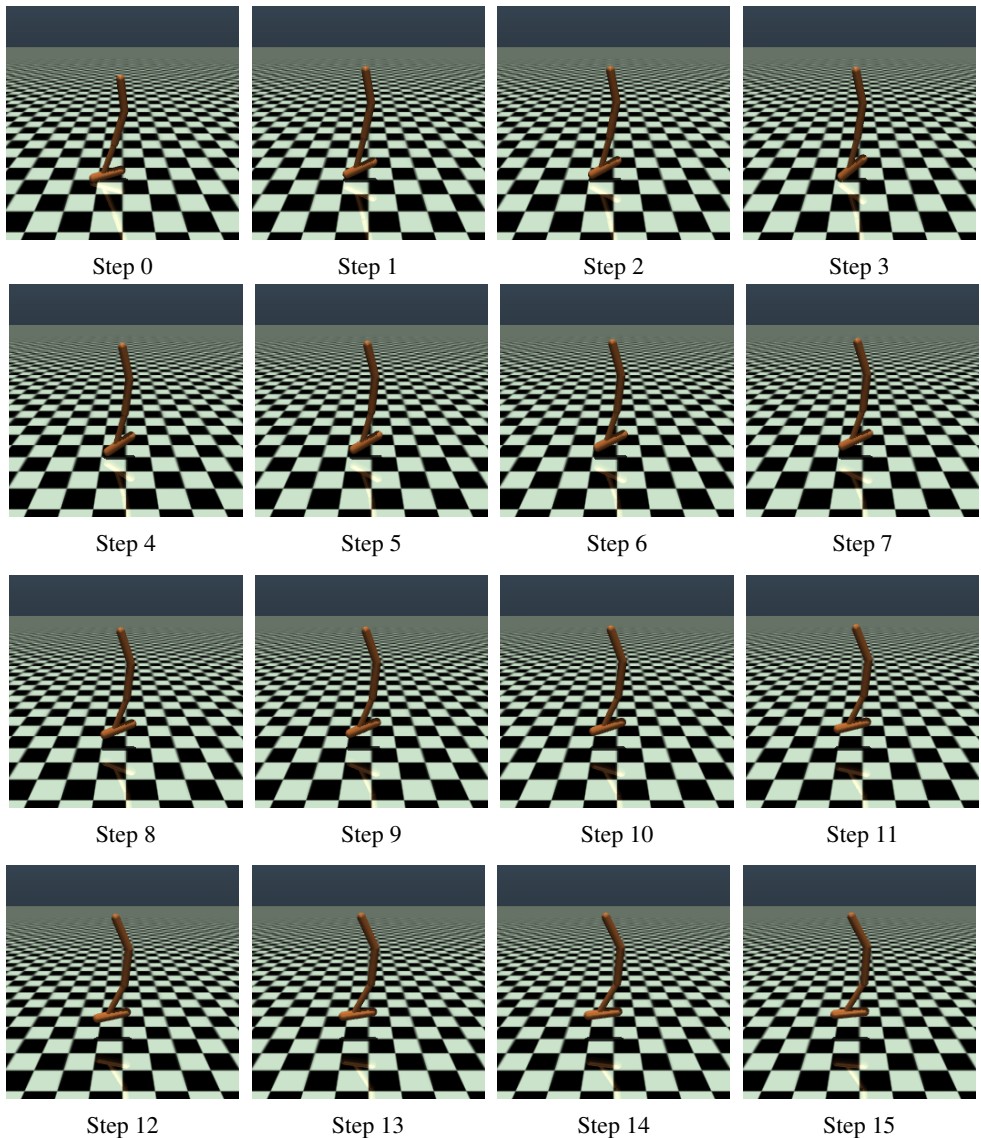

Figure 5: Ensemble MLP

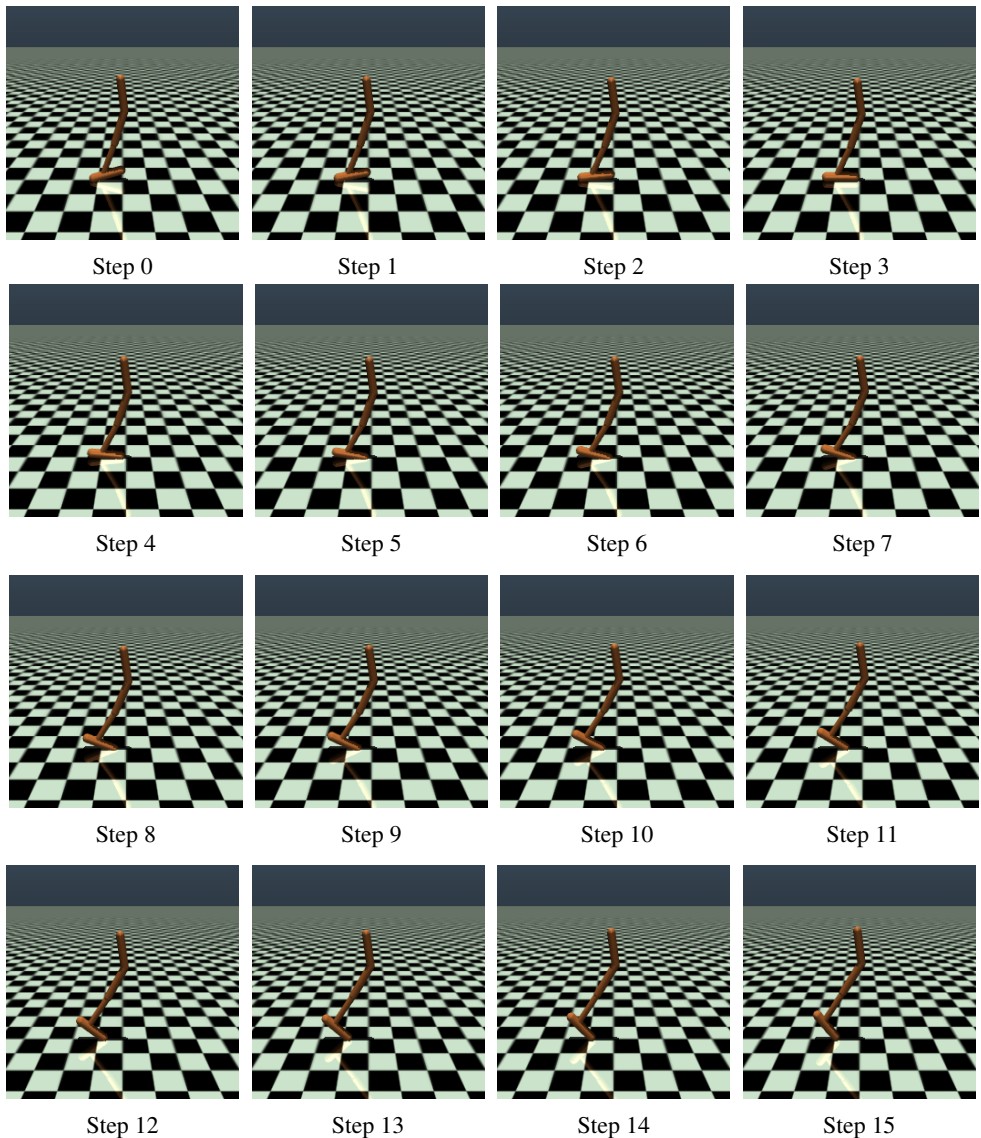

Figure 6: Diffusion

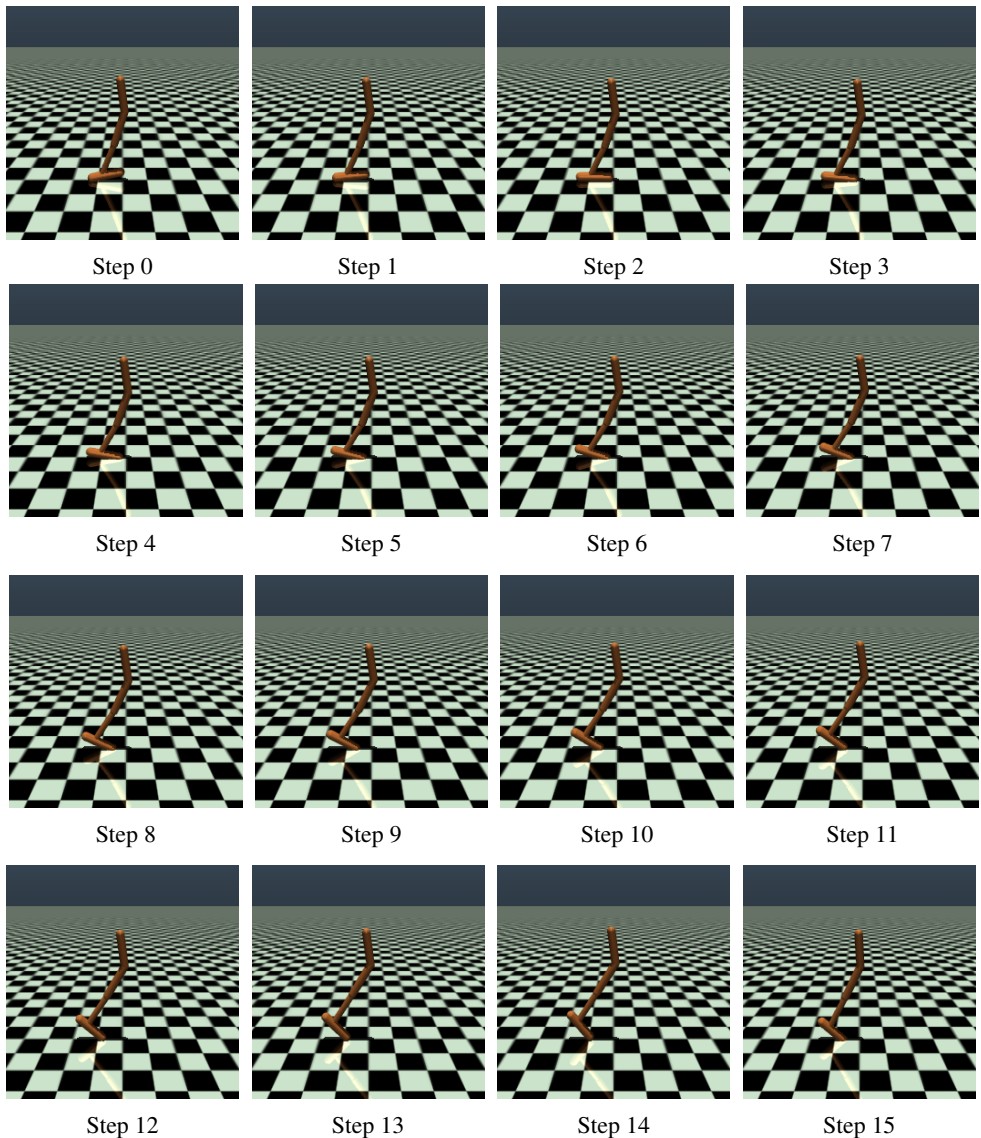

Figure 7: Transformer

| Method | HalfCheetah | | | Hopper | | | Walker2d | | |
|---|---|---|---|---|---|---|---|---|---|
| | 4 | 8 | 16 | 4 | 8 | 16 | 4 | 8 | 16 |
| | | | | | *medium* | | | | |
| Aug-BC | 23.1 ± 1.5 | 5.0 ± 0.9 | 3.6 ± 0.8 | 65.3 ± 1.8 | 52.1 ± 1.6 | 46.5 ± 2.1 | 66.1 ± 1.7 | 51.6 ± 1.9 | 14.0 ± 2.5 |
| Aug-CQL | 24.2 ± 1.4 | 3.8 ± 0.8 | 3.7 ± 0.7 | 67.7 ± 1.5 | 66.2 ± 1.8 | 21.1 ± 2.0 | 75.8 ± 2.3 | 31.2 ± 2.1 | 13.0 ± 2.6 |
| Belief-CQL | 49.2 ± 1.5 | 8.9 ± 1.2 | 3.0 ± 0.7 | 75.4 ± 1.3 | 59.8 ± 2.1 | 42.9 ± 2.6 | 87.0 ± 1.6 | 64.1 ± 2.3 | 39.2 ± 2.8 |
| Belief-IQL | 30.8 ± 1.5 | 10.6 ± 1.0 | 5.3 ± 0.8 | 27.7 ± 1.3 | 29.3 ± 1.4 | 25.4 ± 1.5 | 33.4 ± 1.6 | 25.7 ± 1.4 | 24.6 ± 1.3 |
| **DT-CORL** | 47.4 ± 1.2 | 27.8 ± 1.9 | 6.4 ± 0.8 | 79.4 ± 1.5 | 85.0 ± 1.2 | 71.8 ± 2.2 | 87.4 ± 1.0 | 87.6 ± 1.4 | 86.8 ± 2.3 |
| | | | | | *expert* | | | | |
| Aug-BC | 6.9 ± 0.6 | 5.0 ± 0.5 | 4.2 ± 0.4 | 110.9 ± 0.7 | 103.6 ± 1.0 | 68.7 ± 1.5 | 108.6 ± 0.6 | 95.4 ± 0.9 | 12.1 ± 1.3 |
| Aug-CQL | 3.6 ± 0.4 | 3.7 ± 0.4 | 3.6 ± 0.3 | 112.9 ± 0.4 | 83.5 ± 1.2 | 8.5 ± 0.8 | 108.7 ± 0.5 | 34.8 ± 1.0 | 6.6 ± 0.6 |
| Belief-CQL | 1.5 ± 0.3 | 1.5 ± 0.3 | 1.5 ± 0.2 | 81.1 ± 0.8 | 43.3 ± 1.0 | 45.9 ± 1.1 | 111.1 ± 0.6 | 110.8 ± 0.5 | 97.7 ± 1.2 |
| Belief-IQL | 6.8 ± 0.5 | 4.8 ± 0.5 | 3.6 ± 0.4 | 18.7 ± 0.9 | 17.4 ± 0.8 | 15.9 ± 0.7 | 25.5 ± 1.0 | 19.9 ± 0.9 | 16.7 ± 0.8 |
| **DT-CORL** | 20.6 ± 0.6 | 5.1 ± 0.4 | 5.2 ± 0.3 | 112.9 ± 0.5 | 113.1 ± 0.4 | 112.2 ± 0.4 | 110.9 ± 0.4 | 111.2 ± 0.5 | 110.5 ± 0.5 |
| | | | | | *medium-expert* | | | | |
| Aug-BC | 20.5 ± 1.4 | 5.4 ± 0.9 | 5.1 ± 1.0 | 93.3 ± 1.6 | 89.5 ± 1.7 | 49.2 ± 2.2 | 105.7 ± 1.5 | 53.3 ± 2.2 | 12.2 ± 1.9 |
| Aug-CQL | 7.4 ± 0.9 | 2.8 ± 0.8 | 1.3 ± 0.6 | 101.7 ± 1.0 | 60.9 ± 2.1 | 17.1 ± 1.4 | 84.4 ± 2.2 | 27.7 ± 1.9 | 1.4 ± 0.5 |
| Belief-CQL | 22.7 ± 1.7 | 6.5 ± 1.1 | 1.5 ± 0.4 | 92.9 ± 1.6 | 39.5 ± 1.7 | 35.2 ± 2.0 | 105.8 ± 1.3 | 99.5 ± 1.4 | 51.0 ± 1.6 |
| Belief-IQL | 24.8 ± 1.4 | 6.1 ± 1.1 | 3.3 ± 0.7 | 26.7 ± 1.3 | 26.6 ± 1.3 | 24.7 ± 1.2 | 49.6 ± 1.7 | 17.3 ± 1.2 | 16.4 ± 1.1 |
| **DT-CORL** | 44.7 ± 2.5 | 21.3 ± 2.2 | 8.7 ± 0.9 | 113.0 ± 0.8 | 112.2 ± 0.7 | 109.9 ± 0.9 | 112.1 ± 0.7 | 112.0 ± 0.6 | 118.1 ± 1.1 |
| | | | | | *medium-replay* | | | | |
| Aug-BC | 21.7 ± 1.9 | 4.2 ± 1.0 | 5.2 ± 1.2 | 25.8 ± 2.6 | 28.0 ± 2.4 | 21.7 ± 2.2 | 26.1 ± 2.1 | 13.5 ± 2.5 | 7.5 ± 1.9 |
| Aug-CQL | 9.2 ± 1.5 | 2.0 ± 0.9 | 3.0 ± 1.0 | 85.7 ± 1.8 | 5.1 ± 1.2 | 4.0 ± 1.4 | 48.5 ± 2.3 | 8.0 ± 1.6 | 3.1 ± 1.1 |
| Belief-CQL | 36.1 ± 2.5 | 14.4 ± 2.0 | 6.4 ± 1.5 | 110.1 ± 2.8 | 99.7 ± 2.4 | 96.6 ± 2.1 | 93.3 ± 3.1 | 93.5 ± 2.8 | 61.0 ± 2.5 |
| Belief-IQL | 23.3 ± 1.3 | 13.8 ± 1.1 | 9.7 ± 1.0 | 24.7 ± 1.2 | 25.5 ± 1.1 | 23.4 ± 1.1 | 28.2 ± 1.4 | 20.6 ± 1.2 | 18.3 ± 1.1 |
| **DT-CORL** | 43.6 ± 3.0 | 27.1 ± 2.0 | 7.9 ± 1.0 | 99.4 ± 2.5 | 100.8 ± 1.9 | 100.2 ± 1.5 | 93.6 ± 2.4 | 90.5 ± 1.9 | 88.1 ± 2.0 |

Table 7: Normalized returns (%) on D4RL MuJoCo locomotion tasks with *deterministic* observation delays of 4, 8, and 16 steps. All results are shown as mean ± std over 3 seeds.

| Method | HalfCheetah | | | Hopper | | | Walker2d | | |
|---|---|---|---|---|---|---|---|---|---|
| | 4 | 8 | 16 | 4 | 8 | 16 | 4 | 8 | 16 |
| | | | | | *medium* | | | | |
| Aug-BC | 25.4 ± 1.6 | 5.2 ± 1.0 | 4.0 ± 1.0 | 60.1 ± 2.0 | 52.5 ± 1.8 | 49.1 ± 2.2 | 65.9 ± 1.9 | 49.9 ± 2.0 | 15.0 ± 2.5 |
| Aug-CQL | 23.6 ± 1.5 | 5.1 ± 1.1 | 4.2 ± 1.2 | 67.8 ± 1.7 | 65.8 ± 2.0 | 22.5 ± 2.2 | 73.5 ± 2.3 | 30.8 ± 2.2 | 9.2 ± 2.5 |
| Belief-CQL | 52.6 ± 1.5 | 46.6 ± 2.0 | 16.2 ± 2.4 | 80.8 ± 1.6 | 67.8 ± 2.1 | 74.3 ± 2.6 | 84.8 ± 1.9 | 81.7 ± 1.8 | 78.9 ± 2.2 |
| Belief-IQL | 33.4 ± 1.6 | 31.2 ± 1.5 | 21.6 ± 1.8 | 27.4 ± 1.3 | 27.4 ± 1.3 | 28.4 ± 1.4 | 34.6 ± 1.7 | 37.5 ± 1.6 | 36.7 ± 1.6 |
| **DT-CORL** | 48.2 ± 1.6 | 47.5 ± 2.0 | 38.4 ± 3.1 | 78.5 ± 1.8 | 72.1 ± 1.6 | 79.3 ± 2.2 | 86.8 ± 1.2 | 87.4 ± 1.4 | 87.0 ± 1.9 |
| | | | | | *expert* | | | | |
| Aug-BC | 7.8 ± 0.7 | 5.6 ± 0.6 | 4.7 ± 0.5 | 112.0 ± 0.6 | 104.6 ± 1.0 | 72.4 ± 1.6 | 108.7 ± 0.8 | 98.7 ± 0.9 | 10.2 ± 1.2 |
| Aug-CQL | 3.2 ± 0.4 | 3.9 ± 0.4 | 3.7 ± 0.3 | 112.5 ± 0.5 | 77.5 ± 1.2 | 6.8 ± 0.9 | 108.8 ± 0.6 | 36.5 ± 1.1 | 7.4 ± 0.7 |
| Belief-CQL | 6.5 ± 0.5 | 2.8 ± 0.4 | 1.7 ± 0.3 | 72.3 ± 0.9 | 35.4 ± 1.2 | 20.1 ± 1.4 | 111.1 ± 0.5 | 111.1 ± 0.6 | 109.6 ± 0.8 |
| Belief-IQL | 13.1 ± 0.8 | 9.2 ± 0.7 | 4.3 ± 0.5 | 18.6 ± 0.9 | 16.7 ± 0.8 | 16.9 ± 0.8 | 43.1 ± 1.1 | 27.7 ± 1.0 | 20.7 ± 0.9 |
| **DT-CORL** | 85.1 ± 1.2 | 12.7 ± 1.0 | 5.8 ± 0.4 | 113.2 ± 0.5 | 112.8 ± 0.6 | 113.1 ± 0.6 | 110.9 ± 0.5 | 110.9 ± 0.6 | 110.5 ± 0.5 |
| | | | | | *medium-expert* | | | | |
| Aug-BC | 19.9 ± 1.5 | 5.4 ± 1.0 | 4.8 ± 0.9 | 95.2 ± 1.7 | 91.6 ± 1.8 | 55.5 ± 2.3 | 83.8 ± 1.5 | 54.6 ± 2.2 | 11.4 ± 2.0 |
| Aug-CQL | 6.5 ± 0.8 | 3.3 ± 0.6 | 0.6 ± 0.4 | 112.9 ± 0.6 | 61.3 ± 1.8 | 18.0 ± 1.6 | 82.2 ± 2.3 | 28.8 ± 2.0 | 1.8 ± 0.6 |
| Belief-CQL | 48.0 ± 2.0 | 22.1 ± 1.8 | 3.6 ± 1.0 | 91.8 ± 1.5 | 48.9 ± 1.8 | 57.5 ± 2.2 | 112.1 ± 1.4 | 106.5 ± 1.6 | 85.1 ± 2.0 |
| Belief-IQL | 31.1 ± 1.5 | 16.3 ± 1.2 | 8.2 ± 0.8 | 28.3 ± 1.3 | 27.9 ± 1.3 | 23.8 ± 1.2 | 49.4 ± 1.6 | 25.0 ± 1.3 | 20.4 ± 1.2 |
| **DT-CORL** | 70.0 ± 2.2 | 44.3 ± 2.1 | 31.7 ± 2.8 | 113.6 ± 0.8 | 112.7 ± 0.7 | 85.4 ± 1.5 | 114.1 ± 0.7 | 113.6 ± 0.6 | 111.5 ± 1.0 |
| | | | | | *medium-replay* | | | | |
| Aug-BC | 17.0 ± 2.0 | 4.6 ± 1.0 | 4.4 ± 1.0 | 23.9 ± 2.6 | 27.2 ± 2.5 | 21.7 ± 2.2 | 24.6 ± 2.0 | 14.0 ± 2.5 | 9.1 ± 2.2 |
| Aug-CQL | 9.4 ± 1.6 | 2.3 ± 1.0 | 1.6 ± 0.8 | 92.4 ± 2.0 | 52.9 ± 2.4 | 4.1 ± 1.2 | 41.5 ± 2.4 | 7.1 ± 1.8 | 1.6 ± 1.0 |
| Belief-CQL | 47.1 ± 2.5 | 41.4 ± 2.3 | 19.7 ± 2.2 | 100.8 ± 2.6 | 99.8 ± 2.4 | 99.2 ± 2.2 | 94.9 ± 2.8 | 97.7 ± 2.6 | 95.3 ± 2.4 |
| Belief-IQL | 24.9 ± 1.4 | 20.0 ± 1.2 | 15.3 ± 1.2 | 25.0 ± 1.2 | 24.5 ± 1.1 | 26.0 ± 1.2 | 31.4 ± 1.5 | 25.4 ± 1.3 | 24.2 ± 1.2 |
| **DT-CORL** | 47.7 ± 2.4 | 43.3 ± 2.0 | 30.4 ± 2.1 | 99.4 ± 2.0 | 100.1 ± 1.8 | 98.8 ± 2.0 | 93.0 ± 2.1 | 90.9 ± 1.7 | 91.8 ± 1.9 |

Table 8: Normalized returns (%) on D4RL MuJoCo locomotion tasks with *stochastic* observation delays $\Delta \sim \mathcal{U}(1, k)$, $k \in \{4, 8, 16\}$. Results are shown as mean ± std over 3 seeds.

**Transformer Ablation.** The ablation in Table 9 reveals a clear relationship between the capacity of the transformer belief model and its predictive performance under delayed observations. As the number of parameters decreases both sample efficiency and prediction accuracy degrade substantially. In particular, smaller models (e.g., latent dimension 64 or shallow 4-layer variants) require significantly more epochs to fit the offline trajectories and still exhibit higher prediction error (Performance ↑), indicating difficulty in capturing the temporal dependencies required for belief reconstruction under long delays.

Conversely, larger models (256-dim with 8–10 layers) achieve the best predictive performance while converging in far fewer epochs, demonstrating superior representational power and optimization stability. Notably, the 256×10 model provides the best overall performance and sample efficiency, confirming that a moderately sized transformer is a favorable balance between predictive accuracy and computational cost.

These results highlight a crucial insight: belief estimation under delayed offline RL requires sufficient sequence modeling capacity, as delay-induced temporal misalignment increases the effective horizon and dynamics complexity. Models that are too lightweight fail to capture these dependencies, while overly large models provide diminishing returns relative to their computational cost. Thus, DT-CORL's chosen transformer configuration emerges as the most robust and well-calibrated architecture for belief prediction under realistic delay settings.

Table 9: Ablation on transformer belief model size in `Hopper-medium-v2` with delay $\Delta = 16$. Results averaged over 3 seeds.

| Latent Dim | Layers | Params (M) | Memory (MB) | Epochs Converge $\leq 0.01$) | Performance ($MSE$) |
|---|---|---|---|---|---|
| 256 | 10 | 7.87 | 17.5 | 14.3 | 0.32 |
| 256 | 8 | 7.08 | 14.3 | 18.0 | 0.63 |
| 256 | 6 | 6.28 | 11.1 | 29.0 | 0.95 |
| 256 | 4 | 5.49 | 7.96 | – | 1.24 |
| 128 | 10 | 4.64 | 4.56 | 61.3 | 1.44 |
| 64 | 10 | 3.82 | 1.28 | – | 2.68 |

Table 10: Long-delay robustness (32 and 64 steps) across HalfCheetah-, Hopper-, and Walker2d-medium-v2 tasks. DT-CORL consistently outperforms belief-based baselines in both deterministic and stochastic delay settings.

| Method | HalfCheetah-med-v2 | | Hopper-med-v2 | | Walker2d-med-v2 | |
|---|---|---|---|---|---|---|
| | 32 (Det/Stoch) | 64 (Det/Stoch) | 32 (Det/Stoch) | 64 (Det/Stoch) | 32 (Det/Stoch) | 64 (Det/Stoch) |
| Belief-IQL | 5.14 / 9.31 | 4.92 / 7.17 | 11.1 / 22.6 | 2.70 / 6.26 | 12.0 / 20.9 | 10.5 / 13.9 |
| Belief-CQL | 3.53 / 5.61 | 4.09 / 4.32 | 9.05 / 20.4 | 2.71 / 4.57 | 7.45 / 18.3 | 5.30 / 8.71 |
| DT-CORL | **6.36 / 11.1** | **5.81 / 7.63** | **13.8 / 26.1** | **2.77 / 6.48** | **60.0 / 85.4** | **21.4 / 41.3** |

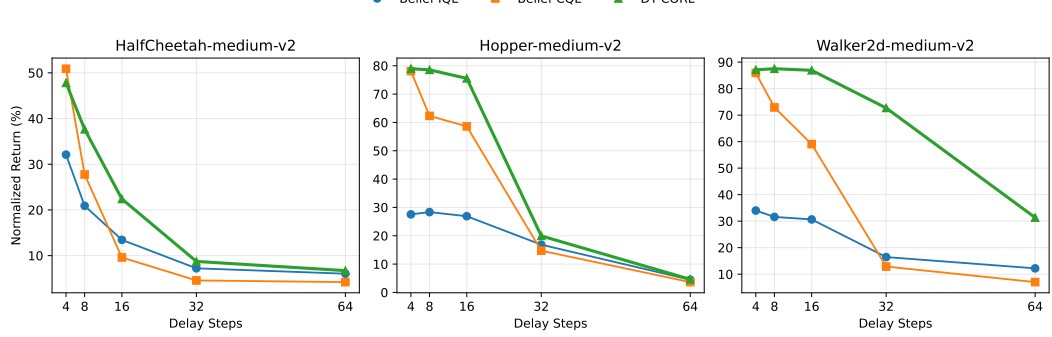

Figure 8: Performance degradation under increasing delay across HalfCheetah-, Hopper-, and Walker2d-medium-v2 tasks. Curves show the average return across deterministic and stochastic delay settings.

# D  IMPLEMENTATION AND REPRODUCIBILITY DETAILS

The implementation of Augmented-CQL, Augmented-BC and our DT-CORL is based on CORL (Tarasov et al., 2023b) and CleanRL (Huang et al., 2022). The implementation of DBT-SAC and AD-SAC can be found in the original paper (Wu et al., 2025; 2024b). And the implementation of Model-based Offline RL(MOPO and COMBO) is based on the OfflineRL-Kit Library (Sun, 2023). We detail the hyperparameter settings of Transformer Belief and DT-CORL in Table 12 and Table 15, respectively. All the experiments are conduct on the server equipped with NVIDIA A5000 GPU and AMD EPYC 7H12 64-Core CPU. The training time depends on state/action dimensions, offline data size, and max delay horizon, which determines the training time of belief predictions. To give an approximate range, training on `Hopper-medium-v2` with 4 delay steps take up around 2 hours, and training on Adroit-Pen task with 16 delay steps take up around 7 hours.

Table 11: Performance on Adroit Hand tasks (Pen-expert-v1, Door-expert-v1, Hammer-expert-v1) under deterministic and stochastic delays $\Delta \in \{4, 8, 16\}$. Best per column is shown in **bold** with light blue background.

| Setting | Method | Pen-expert-v1 | | | Door-expert-v1 | | | Hammer-expert-v1 | | |
|---|---|---|---|---|---|---|---|---|---|---|
| | | **4** | **8** | **16** | **4** | **8** | **16** | **4** | **8** | **16** |
| Deterministic | Aug-CQL | 20.11 | 4.38 | -0.65 | 78.96 | 61.74 | 40.12 | 0.25 | 0.23 | 0.21 |
| | Belief-CQL | 50.43 | 26.57 | 19.43 | 97.61 | 83.35 | 67.91 | 0.27 | 0.29 | 0.22 |
| | **DT-CORL** | **86.44** | **77.51** | **66.38** | **102.57** | **100.65** | **93.04** | **114.51** | **110.12** | **105.20** |
| Stochastic | Aug-CQL | 15.15 | 6.17 | -0.72 | 67.80 | 49.88 | 25.84 | 0.27 | 0.26 | 0.24 |
| | Belief-CQL | 64.68 | 27.42 | 22.51 | 97.59 | 85.92 | 69.73 | 0.28 | 0.29 | 0.24 |
| | **DT-CORL** | **90.56** | **79.33** | **74.85** | **102.97** | **101.47** | **97.43** | **122.13** | **112.22** | **112.33** |

Table 12: Hyper-parameters table of Transformer Belief.

| Hyper-parameter | Value |
|---|---|
| Epoch | 1e2 |
| Batch Size | 256 |
| Attention Heads Num | 4 |
| Layers Num | 10 |
| Hidden Dim | 256 |
| Attention Dropout Rate | 0.1 |
| Residual Dropout Rate | 0.1 |
| Hidden Dropout Rate | 0.1 |
| Learning Rate | 1e-4 |
| Optimizer | AdamW |
| Weight Decay | 1e-4 |
| Betas | (0.9, 0.999) |

Table 13: Hyper-parameters table of Ensemble Belief.

| Hyper-parameter | Value |
|---|---|
| Epoch | 1e2 |
| Batch Size | 256 |
| Attention Heads Num | 4 |
| Layers Num | 10 |
| Hidden Dim | 256 |
| Attention Dropout Rate | 0.1 |
| Residual Dropout Rate | 0.1 |
| Hidden Dropout Rate | 0.1 |
| Learning Rate | 1e-4 |
| Optimizer | AdamW |
| Weight Decay | 1e-4 |
| Betas | (0.9, 0.999) |

Table 14: Hyper-parameters table of Diffusion Belief.

| Parameter | Value |
|---|---|
| Model Dimension | 32 |
| Embedding Dimension | 32 |
| Dimension Multipliers | [1, 2, 2, 2] |
| EMA Rate | 0.9999 |
| Diffusion Steps | 20 |
| Predict Noise | True |
| State Loss Weight | 10.0 |
| Action Loss Weight | 10.0 |
| Classifier Guidance (w_cg) | 0.3 |
| Timestep Embedding | Positional |
| Kernel Size | 5 |
| Solver | DDPM |
| Sampling Steps | 20 |
| Temperature | 1.0 |

Table 15: Hyper-parameters table of DT-CORL.

| Hyper-parameter | Value |
|---|---|
| Learning Rate (Actor) | 3e-4 |
| Learning Rate (Critic) | 1e-3 |
| Learning Rate (Entropy) | 1e-3 |
| Train Frequency (Actor) | 2 |
| Train Frequency (Critic) | 1 |
| Soft Update Factor (Critic) | 5e-3 |
| Batch Size | 256 |
| Neurons | [256, 256] |
| Action Noise | 0.2 |
| Layers | 3 |
| Hidden Dim | 256 |
| Activation | ReLU |
| Optimizer | Adam |

