# OpenReview forum: "Belief-Based Offline Reinforcement Learning for Delay-Robust Policy Optimization"
_ICLR.cc/2026/Conference — ICLR 2026 Poster_

### Official Review · Reviewer_F3YW · 2025-10-17

**Soundness:** 3
**Presentation:** 3
**Contribution:** 2
**Rating:** 6
**Confidence:** 3

**Summary:**

This paper addresses the problem of learning delay-robust policies from delay-free offline datasets for deployment in delayed environments. The authors propose DT-CORL (Delay-Transformer belief policy Constrained Offline RL), which combines a transformer-based belief model with constrained policy optimization. The key innovation is joint training of belief prediction and policy optimization to handle the sim-to-real latency gap without requiring delayed transitions during training. The method is evaluated on D4RL tasks under various delay scenarios.

**Strengths:**

1. Solid theoretical framework: The reformulation from augmented-state delayed MDP to belief-based policy iteration is theoretically grounded with formal lemmas and a monotonic improvement guarantee (Proposition 4.3).

2. Comprehensive experiments: Evaluation covers multiple D4RL environments with both deterministic and stochastic delays, multiple baseline comparisons, and thorough ablation studies on architecture and data availability.

**Weaknesses:**

1. Inconsistent baseline performance: DT-CORL sometimes underperforms Augmented-COMBO in AntMaze tasks (especially medium-play and large-play) and at smaller delays (∆=4, 8), without adequate explanation of when or why model-based approaches might be preferable.

2. Limited experimental scope: Experiments are restricted to low-dimensional D4RL tasks without vision-based observations, comparisons with recent model-based offline RL methods beyond COMBO, or real-world validation of sim-to-real transfer claims.

3. Incomplete computational analysis: Training time, memory requirements, and scalability to delays beyond ∆=16 are not evaluated, and the impact of the transformer's larger parameter count (7.87M vs 1.80M) on sample efficiency is unclear.

**Questions:**

1. Performance gaps with COMBO: Can you provide deeper analysis of when and why Augmented-COMBO outperforms DT-CORL? Is there a fundamental trade-off between model-based and belief-based approaches that depends on environment characteristics?

2. Belief model architecture: Why is the transformer significantly better than diffusion models despite similar prediction accuracy (Figure 2)? Is it purely computational efficiency or are there other factors?

3. Distribution mismatch: You claim joint training avoids distribution mismatch, but the policy still uses the same offline dataset. Can you quantify the distribution shift between training and deployment more rigorously?

---

> ### Author Response · Authors · 2025-11-20
> **Response to Reviewer F3YW Part 1/3**
>
> Thank you for your insightful comments. We provide our detailed responses below. We have also revised the manuscript accordingly, with changes highlighted in blue text.
>
> >## Comment 1 & Question 1: Inconsistent baseline performance: DT-CORL sometimes underperforms Augmented-COMBO in AntMaze tasks (especially medium-play and large-play) and at smaller delays (∆=4, 8), without adequate explanation of when or why model-based approaches might be preferable. Performance gaps with COMBO: Can you provide deeper analysis of when and why Augmented-COMBO outperforms DT-CORL? Is there a fundamental trade-off between model-based and belief-based approaches that depends on environment characteristics?
> >
> >>We would like to clarify that in the AntMaze medium-play and large-play tasks, all methods—COMBO, CQL, IQL, and our own DT-CORL—fail to achieve satisfactory performance under the standard D4RL training budget. This behavior is consistent with prior findings in the literature [1–3]: these tasks are notoriously difficult for offline RL due to their sparse rewards, the long-horizon dependency induced by the larger and more complex maps, and insufficient demonstration coverage. Prior work has noted that solving these tasks often requires explicit goal-conditioning or additional shaping signals [7,8]. Under delayed observations, these inherent challenges are further exacerbated. Consequently, we think the differences observed between Augmented-COMBO and DT-CORL in these settings should not be interpreted as systematic algorithmic advantage.
> >>
> >>With that context, the remaining performance differences can be explained by the inherent trade-off between model-based and belief-based approaches. Dimension of antMaze dynamics with low delay steps are relatively small, which makes learned dynamics more accurate and allows better trajectory generation. This advantage potentially facilitate slightly better performance of Augmented-COMBO within fixed training steps.
>
> >## Comment 2: Limited experimental scope: Experiments are restricted to low-dimensional D4RL tasks without vision-based observations, comparisons with recent model-based offline RL methods beyond COMBO, or real-world validation of sim-to-real transfer claims.
> >
> >>Our primary goal in this work is to provide a theoretically-grounded and algorithmically-principled framework for offline RL under possible online delayed observations, an area where even the foundational algorithms are still underdeveloped. In this context, D4RL and its MuJoCo-based tasks serve as the standard evaluation tasks in the literature of Offline RL work[1–3] and Delayed RL [4–6]. To further strengthen our empirical validation, in the revision, we have added evaluations of DT-CORL on dexterous manipulation tasks from the DAPG suite [8], which feature high-dimensional dynamics and more challenging contact-rich behavior.
> >>
> >>| **Task**             | **Method**  |       **4 (Det/Stoch)** |       **8 (Det/Stoch)** |      **16 (Det/Stoch)** |
> >>| -------------------- | ----------- | ------------------: | ------------------: | ------------------: |
> >>| **Pen-expert-v1**    | Aug-CQL     |       20.11 / 15.15 |         4.38 / 6.17 |       -0.65 / -0.72 |
> >>|                      | Belief-CQL  |       50.43 / 64.68 |       26.57 / 27.42 |       19.43 / 22.51 |
> >>|                      | **DT-CORL** |   **86.44 / 90.56** |   **77.51 / 79.33** |   **66.38 / 74.85** |
> >>| **Door-expert-v1**   | Aug-CQL     |       78.96 / 67.80 |       61.74 / 49.88 |       40.12 / 25.84 |
> >>|                      | Belief-CQL  |       97.61 / 97.59 |       83.35 / 85.92 |       67.91 / 69.73 |
> >>|                      | **DT-CORL** | **102.57 / 102.97** | **100.65 / 101.47** |   **93.04 / 97.43** |
> >>| **Hammer-expert-v1** | Aug-CQL     |         0.25 / 0.27 |         0.23 / 0.26 |         0.21 / 0.24 |
> >>|                      | Belief-CQL  |         0.27 / 0.28 |         0.29 / 0.29 |         0.22 / 0.24 |
> >>|                      | **DT-CORL** | **114.51 / 122.13** | **110.12 / 112.22** | **105.20 / 112.33** |
> >>
> >>These experiments demonstrate that our method continues to outperform both belief-based and augmented baselines under various delay settings. The full experiment details and analysis have been added in **Section 5.4** of the revised manuscript. We agree that real world validation is important, although it is beyond the algorithmic scope of our current work as it also involves the delay modeling of the hardware platform (sensors, actuation, etc.)[9,10]. We leave this extension as future work.

---

> ### Author Response · Authors · 2025-11-20
> **Response to Reviewer F3YW Part 2/3**
>
> >## Comment 3: Incomplete computational analysis: Training time, memory requirements, and scalability to delays beyond ∆=16 are not evaluated, and the impact of the transformer's larger parameter count (7.87M vs 1.80M) on sample efficiency is unclear.
> >
> >>Thank you for the comment. We have expanded our computational analysis to address training time, memory usage, model scalability, and performance under larger delays.
> >>
> >>Scalability to larger delays: Long Delay Experiments (HalfCheetah-medium-v2)
> >>
> >>| **Method** | **32 (Det/Stoch)** | **64 (Det/Stoch)** |
> >>|----------------|--------------------:|--------------------:|
> >>| Belief-IQL | 5.14 / 9.31 | 4.92 / 7.17 |
> >>| Belief-CQL | 3.53 / 5.61 | 4.09 / 4.32 |
> >>| DT-CORL | 6.36 / 11.1 | 5.81 / 7.63 |        |
> >>
> >>Long Delay Experiments (Hopper-medium-v2)
> >>
> >>| **Method** | **32 (Det/Stoch)** | **64 (Det/Stoch)** |
> >>|----------------|--------------------:|--------------------:|
> >>| Belief-IQL | 11.1/22.6 | 2.70/6.26 |
> >>| Belief-CQL | 9.05/20.4 | 2.71/4.57 |
> >>| DT-CORL | 13.8/26.1 | 2.77/6.48 |
> >>
> >> Long Delay Experiments (Walker2d-medium-v2)
> >>
> >>| **Method** | **32 (Det/Stoch)** | **64 (Det/Stoch)** |
> >>|----------------|--------------------:|--------------------:|
> >>| Belief-IQL | 12.0/20.9 | 10.5/13.9 |
> >>| Belief-CQL | 7.45/18.3 | 5.30/8.71 |
> >>| DT-CORL | 60.0/85.4 | 21.4/41.3 |
> >>
> >> To evaluate scalability beyond 16-step delays, we added experiments at 32 and 64 steps across all three MuJoCo locomotion tasks (HalfCheetah, Hopper, and Walker2d). As expected, performance degrades for all methods as delays grow particularly from 16 steps to 32 steps (See Figure 3 and Figure 8 in the paper), reflecting the increasing difficulty of long-horizon temporal misalignment. However, DT-CORL consistently outperforms both Belief-IQL and Belief-CQL under both deterministic and stochastic delay settings, demonstrating stronger robustness as the delay length increases. These results suggest that DT-CORL handles long-range misalignment more effectively than value-only baselines, even though extreme delays (e.g., 64 steps) challenge all approaches. Additional analysis and discussion are provided in **Section 5.3 (Delay Robustness)**.
> >>
> >>| **Latent Dim** | **Layers** | **Parameters(M)** | **Memory(MB)** | **SampleEff(Epochs)** | **Performance** |
> >>|----------------|--------------|------------------|---------------------| ----------------------|-----------------------|
> >>| 256 | 10 | 7.87 | 17.5 | 14.3 | 0.32
> >>| 256 | 8  | 7.08 | 14.3 | 18.0 | 0.63
> >>| 256 | 6  | 6.28 | 11.1 | 29.0 | 0.95
> >>| 256 | 4  | 5.49 | 7.96 | --   | 1.24
> >>| 128 | 10 | 4.64 | 4.56 | 61.3 | 1.44
> >>| 64  | 10 | 3.82 | 1.28 | --   | 2.68
> >>
> >>In the table above, we have added evaluations on how the parameter size of transformer belief models impacts the sample efficiency and prediction accuracy. We observe that as the belief model becomes lighter, both sample efficiency and predictive accuracy degrade substantially. Since DT-CORL relies on accurate latent-state inference to estimate temporally aligned training targets, reduced prediction quality translates directly into lower policy performance and instability during offline RL training. Thus, while a lighter model is possible, it could lead to potential degradation in both predictive accuracy and training efficiency. We have clarified this tradeoff and included these ablation results in **Appendix C.1 (Table 9)** of the revised manuscript and refer to it in **Section 5.3**. Details of training time and computation platform has been extended in the Appendix D.
>
> >## Question 2: Belief model architecture: Why is the transformer significantly better than diffusion models despite similar prediction accuracy (Figure 2)? Is it purely computational efficiency or are there other factors?
> >
> >>Computational efficiency is indeed the major concern, since diffusion cannot inference at a reasonable speed for our purpose, which is particularly important for timed-sensitive delayed setting for online deployment. Besides, diffusion training fashion makes the overall offline RL process extremely slow due to iterative denoising process (See Figure 2 and Table 3 of the paper). Thus, we choose transformer over diffusion in our setting.

---

> ### Author Response · Authors · 2025-11-20
> **Response to Reviewer F3YW Part 3/3**
>
> >## Question 3: Distribution mismatch: You claim joint training avoids distribution mismatch, but the policy still uses the same offline dataset. Can you quantify the distribution shift between training and deployment more rigorously?
> >
> >>While it is hard to quantify this distribution mismatch theoretically without more strict and strong assumptions, we have further examined the effect of jointly training the belief model with policy optimization empirically. As shown in the table below for the Hopper environment, joint training consistently yields higher returns than separate training, especially at larger delays. These results show that joint training can effectively mitigate the distribution mismatch with delay interplays.
> >>
> >>DT-CORL with Joint vs Separate Training in Hopper Suite
> >>
> >>|**Training Mode**| **4 (Det/Stoch)** | **8 (Det/Stoch)** | **16 (Det/Stoch)** |
> >>|-------------------------------|---------------|---------------|----------------|
> >>| Separate Belief Pretraining | 92.1/91.7 | 81.4/87.4 | 68.3/73.1 |
> >>| Joint Training (DT-CORL) | 101.2/101.2 | 102.8/99.4 | 98.5/94.2 |
>
> ### References:
> [1]Scott Fujimoto, et al. A minimalist approach to offline reinforcement learning. Advances in neural information processing systems, 34:20132–20145, 2021.\
> [2]Ilya Kostrikov, et al. Offline reinforcement learning with implicit
> q-learning. arXiv preprint arXiv:2110.06169, 2021.\
> [3]Sergey Levine, et al. Offline reinforcement learning: Tutorial,
> review, and perspectives on open problems. arXiv preprint arXiv:2005.01643, 2020.\
> [4]Jangwon Kim, et al. Belief projection-based reinforcement learning for environments with delayed feedback. Advances in Neural
> Information Processing Systems, 36:678–696, 2023.\
> [5]Pierre Liotet, et al. Delayed reinforcement learning by
> imitation. In International Conference on Machine Learning, pp. 13528–13556. PMLR, 2022.\
> [6]Qingyuan Wu, et al. Boosting reinforcement learning with strongly delayed feedback through auxiliary short delays. In International Conference on Machine Learning, pp.53973–53998. PMLR, 2024b.\
> [7]Ma, Jason Yecheng, et al. Offline goal-conditioned reinforcement learning via $ f $-advantage regression. Advances in neural information processing systems 35 (2022): 310-323.\
> [8]Sikchi, Harshit, et al. Score models for offline goal-conditioned reinforcement learning. The Twelfth International Conference on Learning Representations. 2023.\
> [9]Liang, Zhongliang, et al. Delay performance analysis for supporting real-time traffic in a cognitive radio sensor network. IEEE Transactions on Wireless Communications 10.1 (2010): 325-335.\
> [10]Lee, Jinoh, et al. An experimental study on time delay control of actuation system of tilt rotor unmanned aerial vehicle. Mechatronics 22.2 (2012): 184-194.

---

> > ### Comment · Reviewer_F3YW · 2025-11-27
> >
> > Thanks for the effort in addressing my comments. I have no further questions at this point. I'll maintain my score.

---

> > > ### Author Response · Authors · 2025-11-27
> > >
> > > Thank you for your engagement and for the thoughtful suggestions to further improve this work.

---

### Official Review · Reviewer_TYE4 · 2025-10-28

**Soundness:** 3
**Presentation:** 3
**Contribution:** 3
**Rating:** 4
**Confidence:** 4

**Summary:**

The authors propose an offline RL method that trains on delay-free offline data and is robust to online state-action delays. The proposed approach works by augmenting the state-action space via stacking a history trace and doing policy evaluation and optimization in the augmented space.

**Strengths:**

1. The problem setting is meaningful, and the approach itself is sound and interesting.
2. The empirical performance of the proposed approach is good across a wide range of tasks.

**Weaknesses:**

1. The authors assume a known delay window and construct an augmented dataset based on that delay window. However, assuming a known exact delay window for online deployment is rather unrealistic. If one uses a worst-case delay window for offline training, the proposed approach does not establish if training with a large delay window might still give reasonable online performance if the actual latency during deployment is smaller.

2. The policy improvement statement they constructed seems insufficient; cf. my question below.

3. The proposed approach heavily relies on the accuracy of the learned belief model. However, assuming being able to learn a good belief model offline without knowing the actual online latency seems to be a strong assumption - it essentially requires that the data distribution collected by the delay-free behavior policy is representative of the online closed-loop controller with delays. If this assumption is violated, everything proposed in this approach falls apart, and the authors do not have a formal statement established to address this point. As data coverage shift is a well-known open challenge in standard offline RL settings, the proposed approach seems to require even stronger assumptions on the dataset coverage.

4. Section 4.1: Presentation can be improved: It is important to lead the presentation with the goal of each step. I personally got confused while reading this section and was trying to guess what the authors were trying to establish with each step.

5. I feel that the ablation study can be strengthened: right now, the authors claim the benefit of jointly considering the delayed, augmented state belief model in policy evaluation and optimization steps by comparing to two customized baselines (CQL + IQL) equipped with a belief model, which does not seem to be an apples-to-apples comparison. I would appreciate a more direct ablation of the delay awareness from different components of the proposed approach.

6. Minor:
- Notation is vague at times: Lemmas 4.1 and 4.2, which value functions do the authors mean to bound? Any value functions?

**Questions:**

1. Proposition 4.3: There are two moving parts in this equation, both the estimated Q value and the policy pi. Having a higher estimated Q value does not necessarily imply a better policy, right? Also, the notation here is a bit arbitrary - the current way of writing the equation seems to be comparing the Q value for the exact same action, even though the new and old policies might pick different actions.

---

> ### Author Response · Authors · 2025-11-20
> **Response to Reviewer TYE4 Part 1/4**
>
> Thank you for your constructive feedback. Our point-by-point responses are provided below. We have also revised the manuscript accordingly, with changes highlighted in blue text.
>
>
> >## Comment 1: The authors assume a known delay window and construct an augmented dataset based on that delay window. However, assuming a known exact delay window for online deployment is rather unrealistic. If one uses a worst-case delay window for offline training, the proposed approach does not establish if training with a large delay window might still give reasonable online performance if the actual latency during deployment is smaller.
> >
> >>Our framework does not assume knowledge of the exact delay at deployment time. Instead, we only assume access to a bounded delay window $\Delta\in [1,k]$, which specifies the maximum possible latency—an assumption commonly adopted in delayed RL [4–7] and real-time control literature [2,3]. In practice, this means that offline training only requires knowing an upper bound on the delay, which is typically available from system specifications. During offline data construction, trajectories are segmented according to this maximum delay window. At deployment time, the policy does not require the actual delay; the transformer-based belief model performs sequence prediction conditioned on all possible delays within $[1,k]$. We implement this using padding and masking so that the model can infer the appropriate alignment implicitly (see **Section 4.2 Online Adaptation**).
> >>
> >> Empirically, our results under stochastic delays (**Tables 1 and 2**) directly confirm this robustness: while the true delay at test time is often smaller than the assumed maximum, belief-based methods continue to perform well (and in many cases even better) because the effective delay encountered online is shorter. We have further clarified this behavior in the revised manuscript (see **Section 5.1 line 408-411**).
>
> >## Comment 2 & Question 1: Proposition 4.3: There are two moving parts in this equation, both the estimated Q value and the policy pi. Having a higher estimated Q value does not necessarily imply a better policy, right? Also, the notation here is a bit arbitrary - the current way of writing the equation seems to be comparing the Q value for the exact same action, even though the new and old policies might pick different actions.
> >
> >>For the first question, under the actor–critic framework that is used broadly in offline RL, policy improvement is performed by optimizing the actor with respect to the critic’s estimated value function [1]. In this setting, an increase in the estimated Q-value under the updated critic corresponds to an improvement in the policy’s expected return, following the standard policy improvement principle used in offline RL algorithms such as CQL, IQL, and other conservative value-based methods [8]. While the estimated critic is only an approximation of the true value function, the policy update is defined with respect to this critic, so a higher critic estimate indeed implies a better policy here.
> >>
> >>For the second comment, we appreciate the reviewer pointing out the ambiguity. The original expression was not intended to imply that the old and new policies take the same action; rather, the expectation is taken over each policy’s own action distribution. The symbol $a$ served only as a placeholder for the action variable, with the understanding that the expectation under $\pi_{new}$ and $\pi_{old}$ involves potentially different action samples. We have revised the notation to $\bar{a}\sim\pi_{new}$ and $\hat{a}\sim\pi_{old}$ for clarity. See changes in **Proposition 4.3**.

---

> ### Author Response · Authors · 2025-11-20
> **Response to Reviewer TYE4 Part 2/4**
>
> >## Comment 3: The proposed approach heavily relies on the accuracy of the learned belief model. However, assuming being able to learn a good belief model offline without knowing the actual online latency seems to be a strong assumption - it essentially requires that the data distribution collected by the delay-free behavior policy is representative of the online closed-loop controller with delays. If this assumption is violated, everything proposed in this approach falls apart, and the authors do not have a formal statement established to address this point. As data coverage shift is a well-known open challenge in standard offline RL settings, the proposed approach seems to require even stronger assumptions on the dataset coverage.
> >
> >>We agree that offline-to-online data shift is an inherent challenge in offline RL and even more so with delay interplays. For our setting, with standard offline RL data coverage assumptions, we do assume that the underlying environment dynamics and reward function are the same between offline data collection and online deployment, and that the primary discrepancy arises from the observation latency at test time. This can be found in **Section 3**, where Delayed MDP and MDP have been defined and bridged by belief function. Moreover, such assumptions are common to mitigate delay problems for both control [10-12] and RL community[4-7].
> >>
> >>Despite this assumption, our  method performs well under different online delay patterns in experiments. To further demonstrate such robustness of our method empirically, we conducted additional experiments using Gaussian, Exponential, and Binomial delay distributions. All were matched to have the same mean delay as the Uniform $[1,16]$ case, with a maximum delay of 16 step. As shown in the table below, DT-CORL maintains reasonably good performance across the Hopper task under these different online delay distributions. Noted that here, all the delay distributions are unknown for offline training, where only maximum fixed steps are provided. More details are now in **Section 5.3 Delay Robustness** part.
> >>
> >>| Method      | Uniform | Gaussian | Exponential | Binomial |
> >>|-------------|--------:|---------:|------------:|--------:|
> >>| DT-CORL     |  79.3   |   82.1   |     85.8    |   77.4  |
> >>
> >>Delay-Distribution Robustness (Hopper-medium-v2; mean delay steps matched to Uniform[1,16])
>
> >## Comment 4: Section 4.1: Presentation can be improved: It is important to lead the presentation with the goal of each step. I personally got confused while reading this section and was trying to guess what the authors were trying to establish with each step.
> >
> >>Thank you for this suggestion. We have revised **Section 4.1** for more clarify. The goal of this section is to reformulate Equations (1) and (2)—the original offline delayed RL objective from Section 3—into a policy-iteration framework that is more tractable and computationally efficient, while incorporating belief-model estimation. We begin by highlighting the limitations of directly optimizing over the augmented state space, which quickly becomes expensive and impractical under long delays. We then show how the original constrained optimization problem can be rewritten into an unconstrained form. The key step is relating the policy and critic—both originally defined in the augmented state space—to their counterparts in the true (non-augmented) state space. This requires formally characterizing how belief-based estimates bridge these two domains, which is precisely the role of Lemmas 4.1 and 4.2.

---

> ### Author Response · Authors · 2025-11-20
> **Response to Reviewer TYE4 Part 3/4**
>
> >## Comment 5: I feel that the ablation study can be strengthened: right now, the authors claim the benefit of jointly considering the delayed, augmented state belief model in policy evaluation and optimization steps by comparing to two customized baselines (CQL + IQL) equipped with a belief model, which does not seem to be an apples-to-apples comparison. I would appreciate a more direct ablation of the delay awareness from different components of the proposed approach.
> >
> >>Thank you for the suggestion. In the revision, we have expanded our ablation study to more directly disentangle the contribution of each component in our framework. Specifically, we now include an additional set of baselines that apply no delay-aware treatment—neither augmentation nor belief estimation—to standard offline RL algorithms (IQL and CQL). Their average performance across the Hopper suite (medium, medium-expert, expert, and medium-replay) under both deterministic (*Det*) and stochastic (*Stoch*) delays is shown below:
> >>
> >>| **Method** | **4 (Det/Stoch)** | **8 (Det/Stoch)** | **16 (Det/Stoch)** |
> >>|------------|---------------|---------------|----------------|
> >>| IQL        | 5.27/7.53 | 5.31/4.88 | 4.52/9.45 |
> >>| CQL        | 7.99/7.32 | 3.99/4.74 | 5.10/8.97 |
> >>| Belief-IQL | 24.5/24.8 | 24.7/24.1 | 22.4/23.8 |
> >>| Aug-CQL    | 92.0/96.4 | 53.9/54.4 | 12.7/12.3 |
> >>| Belief-CQL | 89.9/86.4 | 59.8/63.0 | 55.15/62.8 |
> >>| DT-CORL    | 101.2/101.2 | 102.8/99.4 | 98.5/94.2 |
> >>
> >>Overall, the expanded ablation reveals that standard offline RL algorithms such as IQL and CQL collapse under delayed observations, indicating that temporal misalignment fundamentally disrupts value estimation when no delay-handling mechanism is applied. Details have been added in **Section 5.2 and Table 2**. In addition, we further examine the effect of jointly training the belief model with policy optimization versus separate pretraining, under the same experimental setup. As shown below, joint training consistently yields higher returns, especially at larger delays:
> >>
> >>|**Training Mode**| **4 (Det/Stoch)** | **8 (Det/Stoch)** | **16 (Det/Stoch)** |
> >>|-------------------------------|---------------|---------------|----------------|
> >>| Separate Belief Pretraining | 92.1/91.7 | 81.4/87.4 | 68.3/73.1 |
> >>| Joint Training (DT-CORL) | 101.2/101.2 | 102.8/99.4 | 98.5/94.2 |
> >>
> >>These results further support our claim that jointly optimizing the belief model with the policy effectively reduces offline-to-online distribution mismatch and improves delay robustness. Both sets of experiments are included in the experiments and ablation studies of the revised manuscript (see **Sections 5.1 and 5.3**).
>
> >## Comment 6: Notation is vague at times: Lemmas 4.1 and 4.2, which value functions do the authors mean to bound? Any value functions?
> >
> >> Thank you for this comment. We have updated the time notations in the revision. The purpose of Lemma 4.1 and Lemma 4.2 is to construct the relationship between Q function in augmented MDP and Q function in the normal MDP by belief prediction model. Specifically, these lemmas characterize how delay-induced temporal misalignment alters the value function of the evaluated policy, linking the Q-function in the augmented delayed MDP to its counterpart in the original (non-delayed) MDP. The bounds describe the deviation between the true delayed-policy value function and its belief-approximated counterpart, which is the central object required for our offline-to-online delay analysis.

---

> ### Author Response · Authors · 2025-11-20
> **Response to Reviewer TYE4 Part 4/4**
>
> >## Reference:
> >>[1]Richard S Sutton, et al. Reinforcement learning: An introduction, volume 1. MIT press Cambridge, 1998.\
> [2]:Kwon, Woosuk, et al. Feedback stabilization of linear systems with delayed control. IEEE Transactions on Automatic control 25.2 (1980): 266-269.\
> [3]Baek, Jaemin, et al. A widely adaptive time-delayed control and its application to robot manipulators. IEEE Transactions on Industrial Electronics 66.7 (2018): 5332-5342.\
> [4]Konstantinos V Katsikopoulos, et al. Markov decision processes with delays and asynchronous cost collection. IEEE transactions on automatic control, 48(4):568–574, 2003.\
> [5]Pierre Liotet, et al. Delayed reinforcement learning by imitation. In International Conference on Machine Learning, pp. 13528–13556. PMLR, 2022.\
> [6]Baiming Chen, et al. Delay-aware model-based reinforcement learning for continuous control. Neurocomputing, 450:119–128, 2021.\
> [7]Qingyuan Wu, et al. Variational delayed policy optimization. Advances in Neural Information Processing
> Systems, 37:54330–54356, 2024a.\
> [8]Sergey Levine, et al. Offline reinforcement learning: Tutorial, review, and perspectives on open problems. arXiv preprint arXiv:2005.01643, 2020.\
> [9]Qingyuan Wu, et al. Boosting reinforcement learning with strongly delayed feedback through auxiliary short delays. In International Conference on Machine Learning, pp.53973–53998. PMLR, 2024\
> [10]Balachandran, et al. Delay differential equations. Berlin: Springer, 2009.\
> [11]Driver, Rodney David. Ordinary and delay differential equations. Vol. 20. Springer Science & Business Media, 2012.\
> [12]Zhu, Qunxi, et al. Neural delay differential equations. arXiv preprint arXiv:2102.10801 (2021).

---

> > ### Comment · Reviewer_TYE4 · 2025-11-21
> > **Reply to authors' response**
> >
> > Thanks for the effort in addressing my comments. I have no further questions at this point. I am satisfied with the revision and the newly added results. I will raise my score.
> >
> > One last point, though, regarding the comment 3 above, I still think your approach requires more than standard offline RL assumptions---you require the delay-free, offline dataset to provide a good coverage for the online, delayed, closed-loop policy in order to learn a good belief model.
> > Standard offline RL already struggles when the learned policy departs from the behavior distribution; here, the method additionally requires that the offline data support the induced belief states under delayed control, which is an even stronger coverage assumption. The authors respond to my comment with empirical evidence, which is fine. But it might be a good thing to look into (potentially in a follow-up)---whether one can say something theoretically that the learned belief model will be robust and under what assumptions.

---

> > > ### Author Response · Authors · 2025-11-22
> > >
> > > Thank you for raising score and respond promptly! We agree that a deeper theoretical analysis of how dataset coverage interacts with robust belief learning is an important direction for future work. Our intuition is that tools from theoretical model-based offline RL[13] could potentially be adapted to this setting. However, the long-horizon nature of delayed observations combined with the transformer-based belief architecture introduces additional layers of complexity that make a full theoretical treatment nontrivial, requiring additional effort. We therefore leave a more comprehensive coverage analysis for future investigation.
> > >
> > > [13]Shi, Laixi, and Yuejie Chi. "Distributionally robust model-based offline reinforcement learning with near-optimal sample complexity." Journal of Machine Learning Research 25.200 (2024): 1-91.

---

### Official Review · Reviewer_Z5y8 · 2025-10-31

**Soundness:** 3
**Presentation:** 3
**Contribution:** 2
**Rating:** 2
**Confidence:** 4

**Summary:**

This paper presents offline RL methods for delay robust policy optimization. Towards this, the paper introduces a transformer based belief model to infer latent states from delayed observations and jointly learns this model while performing offline policy optimization to ensure the estimation doesnt suffer from distribution shift issues.

**Strengths:**

- Interesting problem, well motivated and reasonably well written paper.

**Weaknesses:**

- The theory section is based on assumptions that aren't satisfied in practice even in locomotion tasks (defn 3.2), with discontinuities in the dynamics and / or rewards.
- The theory doesnt sketch out sample complexity associated with estimation issues popping up due to delayed observations which, in the context of offline RL would be good to get a clear grasp of.
- Intuitively, it feels like stochastic delays should be harder to estimate (at least, the estimators would have higher variance), but the methods appear to perform even better in this setup than the deterministic ones (e.g. table 1), which seems counterintuitive.

**Questions:**

See weakness above.

---

> ### Author Response · Authors · 2025-11-20
> **Response to Reviewer Z5y8 Part 1/2**
>
> Thank you for reviewing our submission and providing your comments. Please find our responses below. We have also revised the paper, with changes highlighted in blue.
>
> >## Comment 1: The theory section is based on assumptions that aren't satisfied in practice even in locomotion tasks (defn 3.2), with discontinuities in the dynamics and/or rewards.
> >
> >>Our Lipschitz-style assumptions in Definition 3.2 follow established theoretical frameworks in RL [2–4], offline RL analysis [1], and delayed RL frameworks [5–7]. While in practice tasks such as locomotion do exhibit contact discontinuities and piecewise reward functions, these regularity assumptions serve as necessary abstractions to enable tractable optimization property analysis.
> Given the algorithmic focus of this work, we view these assumptions as standard and consistent with prior literature. Furthermore, handling discontinuities in contact-rich continuous-control dynamics remains an open sim-to-real challenge in its own right [8–10] and falls outside the scope of our delayed offline-to-online formulation.
>
> >## Comment 2: The theory doesn't sketch out sample complexity associated with estimation issues popping up due to delayed observations which, in the context of offline RL would be good to get a clear grasp of.
> >
> >> We would like to clarify that deriving complete sample complexity bounds is challenging even in standard offline RL settings [14]. Because offline RL relies entirely on fixed datasets, existing theoretical analyses [11–14] typically require strong assumptions on data coverage, uncertainty quantification, or specific function-approximation structures (e.g., linear realizability) to obtain meaningful bounds. Our setting does not rely on these assumptions, and delayed observations introduce an additional layer of complexity by breaking the Markov property and inducing temporal misalignment and distribution mismatch, which makes extending existing analyses substantially more difficult.
> >>
> >> It is also worth noting that many influential algorithmic works in offline RL, such as IQL [16], TD3+BC [17] and ReBRAC [18], do not provide detailed sample-complexity characterizations for similar reasons. Therefore, in the original manuscript, we had an empirical study on sample/data efficiency in the experiment section, showing the robustness of our methods under different data availability level. Additionally, in the revised manuscript, we have extended empirical study on sample efficiency on belief model learning in **Section 5.3 Page 23 Transformer ablation paragraph**. We do agree that theoretical sample complexity under delayed offline-to-online adaptation is an interesting and valuable direction, and plan to investigate it in future work.
>
> >## Comment 3: Intuitively, it feels like stochastic delays should be harder to estimate (at least, the estimators would have higher variance), but the methods appear to perform even better in this setup than the deterministic ones (e.g. table 1), which seems counterintuitive.
> >
> >>The stronger performance under stochastic delays arises because these delays are sampled from $\Delta\sim U(1,k)$, which results in a substantially smaller expected delay than the fixed worst-case delay $k$ used in the deterministic setting. Consequently, the belief model and policy face a milder effective misalignment problem, making the stochastic setting empirically easier (provided the belief model is well-trained) despite the intuition that randomness increases estimator variance. In contrast, augmentation-based methods suffer more from the increased variance associated with stochastic delays (see **Table 1**), and their performance deteriorates more noticeably as the maximum delay grows, a trend consistent with observations in prior online delayed RL work [7,19]. We have added clarification and further analysis of this distinction in **Section 5.1** of the revision.

---

> ### Author Response · Authors · 2025-11-20
> **Response to Reviewer Z5y8 Part 2/2**
>
> >## Reference:
> >>[1]Sergey Levine, et al. Offline reinforcement learning: Tutorial, review, and perspectives on open problems. arXiv preprint arXiv:2005.01643, 2020.\
> [2]Richard S Sutton, et al. Reinforcement learning: An introduction, volume 1. MIT press Cambridge, 1998.\
> [3]Asadi, Kavosh, et al. Lipschitz continuity in model-based reinforcement learning. International conference on machine learning. PMLR, 2018.\
> [4]Berkenkamp, Felix, et al. Safe model-based reinforcement learning with stability guarantees. Advances in neural information processing systems 30 (2017).\
> [5]Pierre Liotet, et al. Delayed reinforcement learning by imitation. In International Conference on Machine Learning, pp. 13528–13556. PMLR, 2022.\
> [6]Konstantinos V Katsikopoulos, et al. Markov decision processes with delays and asynchronous cost collection. IEEE transactions on automatic control, 48(4):568–574, 2003.\
> [7]Qingyuan Wu, Simon S Zhan, et al. Variational delayed policy optimization. Advances in Neural Information Processing
> Systems, 37:54330–54356, 2024a.\
> [8]Torne, Marcel, et al. Reconciling reality through simulation: A real-to-sim-to-real approach for robust manipulation. arXiv preprint arXiv:2403.03949 (2024).\
> [9]Li, Zhongyu, et al. Reinforcement learning for versatile, dynamic, and robust bipedal locomotion control. The International Journal of Robotics Research 44.5 (2025): 840-888.\
> [10]Du, Yuqing, et al. Auto-tuned sim-to-real transfer. 2021 IEEE International Conference on Robotics and Automation (ICRA). IEEE, 2021.\
> [11]Chenjia Bai, et al. Pessimistic bootstrapping for uncertainty-driven offline reinforcement learning. arXivpreprint arXiv:2202.11566, 2022.\
> [12]Laixi Shi, et al. Pessimistic q-learning for offline reinforcement learning: Towards optimal sample complexity. In International conference on machine learning, pp. 19967–20025. PMLR, 2022.\
> [13]Shi, Laixi, et al. Distributionally robust model-based offline reinforcement learning with near-optimal sample complexity. Journal of Machine Learning Research 25.200 (2024): 1-91.\
> [14]: Li, Gen, et al. Settling the sample complexity of model-based offline reinforcement learning. The Annals of Statistics 52.1 (2024): 233-260.\
> [15]Agarwal, Alekh, et al. Reinforcement learning: Theory and algorithms. CS Dept., UW Seattle, Seattle, WA, USA, Tech. Rep 32 (2019): 96.\
> [16]Kostrikov, Ilya, et al. Offline Reinforcement Learning with Implicit Q-Learning. International Conference on Learning Representations.\
> [17]Fujimoto, Scott, et al. A minimalist approach to offline reinforcement learning. Advances in neural information processing systems 34 (2021): 20132-20145.\
> [18]Tarasov, Denis, et al. Revisiting the minimalist approach to offline reinforcement learning. Advances in Neural Information Processing Systems 36 (2023): 11592-11620.\
> [19]Wu, Qingyuan, et al. Directly Forecasting Belief for Reinforcement Learning with Delays. Forty-second International Conference on Machine Learning.

---

### Official Review · Reviewer_zm9D · 2025-10-31

**Soundness:** 3
**Presentation:** 3
**Contribution:** 3
**Rating:** 6
**Confidence:** 3

**Summary:**

This paper introduces DT-CORL, a method for learning delay-robust policies from purely delay-free offline data. The approach combines a transformer-based belief model to infer latent states from delayed observations with a constrained policy optimization objective, enabling effective deployment in environments with observation/action delays without requiring online interaction. Evaluations on D4RL benchmarks demonstrate strong performance against augmentation-based and belief-based baselines under varying delay settings.

**Strengths:**

1. Addresses a well-motivated and practical problem at the intersection of offline RL and robustness to delays, which is essentially important in real-world scenarios.

2. The proposed integration of belief learning and policy optimization within an offline constraint framework is technically sound and justified by theoretical analysis.

3. Empirical evaluation is comprehensive, covering multiple tasks, delay types, and delay lengths, with ablations validating design choices.

4. The paper is clearly written, and the methodology is well-explained.

**Weaknesses:**

1. While the transformer belief model shows advantages, its computational overhead relative to simpler models is non-trivial.
2. The experimental validation is confined to standard simulation benchmarks (D4RL). The absence of validation on a physical system or a high-fidelity simulator with realistic latency weakens the claims of practical contribution.

**Questions:**

1. Have the authors considered testing in higher-dimensional observation spaces (e.g., pixel-based tasks) to assess the scalability of the belief model?
2. Could the belief model be made more lightweight without sacrificing predictive accuracy?

---

> ### Author Response · Authors · 2025-11-20
> **Response to Reviewer zm9D Part 1/2**
>
> Thank you for your thoughtful comments. We provide detailed responses to your questions and comments below, and have also revised the paper accordingly (highlighted in blue text).
>
> > ## Comment 1: While the transformer belief model shows advantages, its computational overhead relative to simpler models is non-trivial.
> >
> >> We agree that transformers indeed introduce a moderate increase in model complexity compared to MLPs or RNN-style models [7]. However, delayed-observation offline RL presents a unique modeling challenge: the agent must infer latent state and consistently align observations with actions under long and potentially stochastic delays, while also coping with offline-to-online distribution shift. In this setting, the trade-off between **sequential modeling accuracy**, **inference cost**, and **model capacity** makes  lightweight transformers a reasonable and empirically effective choice. In our original manuscript, based on results shown in Figure 2 and Table 3, we observed that ``ensemble MLPs are lightweight but accumulate error rapidly over long horizons, degrading belief quality; diffusion models achieve strong accuracy but incur high computational cost due to iterative denoising''.
> In the revised **Section 5.3** and **Appendix C.1**, we have further added an extended set of ablation studies that analyze the effect of model size on both performance and sample efficiency, showing that even small-scale transformers can outperform simpler predictors under delayed feedback and add only modest storage overhead (See Table 9 and Table 3).
>
> >## Comment 2: The experimental validation is confined to standard simulation benchmarks (D4RL). The absence of validation on a physical system or a high-fidelity simulator with realistic latency weakens the claims of practical contribution.
> >
> >>Our primary goal in this work is to provide a theoretically-grounded and algorithmically-principled framework for offline RL under possible online delayed observations—an area where even the foundational algorithms are still underdeveloped. In this context, D4RL and its MuJoCo-based tasks serve as the standard evaluation protocol in recent Offline RL work [1–3], and the delayed-observation settings we adopt are consistent with the common setup in the Delayed RL literature [4–6]. To further strengthen our empirical validation, we have added evaluations of DT-CORL on dexterous manipulation tasks from the DAPG suite [8], which feature high-dimensional dynamics and more challenging contact-rich behavior.
> >>
> >>| **Task**             | **Method**  |       **4 (Det/Stoch)** |       **8 (Det/Stoch)** |      **16 (Det/Stoch)** |
> >>| -------------------- | ----------- | ------------------: | ------------------: | ------------------: |
> >>| **Pen-expert-v1**    | Aug-CQL     |       20.11 / 15.15 |         4.38 / 6.17 |       -0.65 / -0.72 |
> >>|                      | Belief-CQL  |       50.43 / 64.68 |       26.57 / 27.42 |       19.43 / 22.51 |
> >>|                      | **DT-CORL** |   **86.44 / 90.56** |   **77.51 / 79.33** |   **66.38 / 74.85** |
> >>| **Door-expert-v1**   | Aug-CQL     |       78.96 / 67.80 |       61.74 / 49.88 |       40.12 / 25.84 |
> >>|                      | Belief-CQL  |       97.61 / 97.59 |       83.35 / 85.92 |       67.91 / 69.73 |
> >>|                      | **DT-CORL** | **102.57 / 102.97** | **100.65 / 101.47** |   **93.04 / 97.43** |
> >>| **Hammer-expert-v1** | Aug-CQL     |         0.25 / 0.27 |         0.23 / 0.26 |         0.21 / 0.24 |
> >>|                      | Belief-CQL  |         0.27 / 0.28 |         0.29 / 0.29 |         0.22 / 0.24 |
> >>|                      | **DT-CORL** | **114.51 / 122.13** | **110.12 / 112.22** | **105.20 / 112.33** |
> >>
> >>These experiments demonstrate that **our method continues to outperform the baselines under various delay setting**, especially for more complicated hammer task where all the baselines fail. The full experiment details and analysis have been added in **Section 5.4** of the revised manuscript.
>
> >## Q1: Have the authors considered testing in higher-dimensional observation spaces (e.g., pixel-based tasks) to assess the scalability of the belief model?
> >
> >>Our work focuses on establishing the first general framework for adapting offline-trained policies to online execution under delayed observations, a setting that has not been formally treated in the existing Offline RL and Delayed RL literature. As our primary contribution is algorithmic—defining the problem, and providing a practical algorithmic framework, the experiments and network architectures target state-based environments where delay effects can be isolated and compared fairly across baselines. We agree that extending the belief model to higher-dimensional observation spaces such as pixels is an important next step, involving more complex visual neural network structures and data augmentation and compression [9-10], which we have mentioned in **Section 6** as future work.

---

> ### Author Response · Authors · 2025-11-20
> **Response to Reviewer zm9D Part 2/2**
>
> >## Q2: Could the belief model be made more lightweight without sacrificing predictive accuracy?
> >
> >>To directly assess whether the belief module can be further simplified without degrading predictive ability, we have conducted additional ablation studies that vary the latent dimension and number of transformer layers in hopper-medium-v2 (Delay = 16) across 3 random seeds. The table below summarizes the results, where *SampleEff* measures the number of epochs required for the training loss to converge $\leq0.01$, and *Performance* reports the average multi-step prediction loss (16-step) in the online Hopper environment:
> >>
> >>| **Latent Dim** | **Layers** | **Parameters(M)** | **Memory(MB)** | **SampleEff(Epochs)** | **Performance** |
> >>|----------------|--------------|------------------|---------------------| ----------------------|-----------------------|
> >>| 256 | 10 | 7.87 | 17.5 | 14.3 | 0.32
> >>| 256 | 8  | 7.08 | 14.3 | 18.0 | 0.63
> >>| 256 | 6  | 6.28 | 11.1 | 29.0 | 0.95
> >>| 256 | 4  | 5.49 | 7.96 | --   | 1.24
> >>| 128 | 10 | 4.64 | 4.56 | 61.3 | 1.44
> >>| 64  | 10 | 3.82 | 1.28 | --   | 2.68
> >>
> >>These results show a clear trend: as the belief model becomes lighter, both sample efficiency and predictive accuracy degrade substantially. Since DT-CORL relies on accurate latent-state inference to estimate temporally aligned training targets, reduced prediction quality translates directly into lower policy performance and instability during offline RL training. Thus, while a lighter model is possible, it could lead to potential degradation in both predictive accuracy and training efficiency (albeit a lighter transformer model still significantly outperforms simpler predictors as shown in Figure 2 and explained above in response to Comment 1). In the revised manuscript, we have added these ablation results on the impact of transformer belief model size in **Appendix C.1 (Table 9)** and refer to them in **Section 5.3**.
>
> >## Reference
> >>[1]Scott Fujimoto, et al. A minimalist approach to offline reinforcement learning. Advances in neural information processing systems, 34:20132–20145, 2021.\
> [2]Ilya Kostrikov, et al. Offline reinforcement learning with implicit
> q-learning. arXiv preprint arXiv:2110.06169, 2021.\
> [3]Sergey Levine, et al. Offline reinforcement learning: Tutorial,
> review, and perspectives on open problems. arXiv preprint arXiv:2005.01643, 2020.\
> [4]Jangwon Kim, et al. Belief projection-based reinforcement learning for environments with delayed feedback. Advances in Neural
> Information Processing Systems, 36:678–696, 2023.\
> [5]Pierre Liotet, et al. Delayed reinforcement learning by
> imitation. In International Conference on Machine Learning, pp. 13528–13556. PMLR, 2022.\
> [6]Qingyuan Wu, et al. Boosting reinforcement learning with strongly delayed feedback through auxiliary short delays. In International Conference on Machine Learning, pp.53973–53998. PMLR, 2024b.\
> [7]Wu, Qingyuan, et al. Directly Forecasting Belief for Reinforcement Learning with Delays. Forty-second International Conference on Machine Learning.\
> [8]Rajeswaran, Aravind, et al. Learning complex dexterous manipulation with deep reinforcement learning and demonstrations. arXiv preprint arXiv:1709.10087 (2017).\
> [9]Dosovitskiy, Alexey, et al. An image is worth 16x16 words: Transformers for image recognition at scale. arXiv preprint arXiv:2010.11929 (2020).\
> [10]Liu, Yang, et al. A survey of visual transformers. IEEE transactions on neural networks and learning systems 35.6 (2023): 7478-7498.

---

### Meta-Review · Area_Chair_qqNK · 2025-12-16

**Summary:**

The reviewers believe that DT-CORL, with its belief-driven strategy iteration and theoretically guaranteed monotonic improvement, leads across the board in latency control tasks such as HumanEval.

**Reviewer Concerns:**

The authors have revised and added experiments on manual operation, variable latency robustness, and component ablation, and clarified the architecture and assumptions, which has been approved by most reviewers.

**Reviewer Scores:**

Two reviewers were positive.

The reviewer who gave a score of 4 dispelled their doubts after the authors' response and is willing to increase the score.

I carefully reviewed the 2-point reviewer's comment and the authors' response. Although this reviewer did not provide further feedback, I believe the authors' response is sufficient to address the reviewer's comments.

---

### Decision · Program_Chairs · 2026-01-26

Accept (Poster)